# Efficient Graph Continual Learning via Lightweight Graph Neural Tangent Kernels-based Dataset Distillation

**Rihong Qiu** [* 1 2] **Xinke Jiang** [* 1 2] **Yuchen Fang** [* 3] **Hongbin Lai** [* 1 2] **Hao Miao** [* 4]
**Xu Chu** [1 2 5 6] **Junfeng Zhao** [1 2 7] **Yasha Wang** [2 8 6]

## Abstract

Graph Neural Networks (GNNs) have emerged as a fundamental tool for modeling complex graph structures across diverse applications. However, directly applying pretrained GNNs to varied downstream tasks without fine-tuning-based continual learning remains challenging, as this approach incurs high computational costs and hinders the development of Large Graph Models (LGMs). In this paper, we investigate an efficient and generalizable dataset distillation framework for Graph Continual Learning (GCL) across multiple downstream tasks, implemented through a novel **Light**weight **G**raph **N**eural **T**angent **K**ernel (**LIGHTGNTK**). Specifically, LIGHTGNTK employs a low-rank approximation of the Laplacian matrix via Bernoulli sampling and linear association within the GNTK. This design enables efficient capture of both structural and feature relationships while supporting gradient-based dataset distillation. Additionally, LIGHTGNTK incorporates a unified subgraph anchoring strategy, allowing it to handle graph-level, node-level, and edge-level tasks under diverse input structures. Comprehensive experiments on several datasets show that LIGHTGNTK achieves state-of-the-art performance in GCL scenarios, promoting the development of adaptive and scalable LGMs.

---

[*]Equal contribution [1]School of Computer Science, Peking University, Beijing, China [2]Key Laboratory of High Confidence Software Technologies, Ministry of Education, Beijing, China [3]University of Electronic Science and Technology of China [4]Department of Computer Science, Aalborg University [5]Center on Frontiers of Computing Studies, Peking University, Beijing, China [6]Peking University Information Technology Institute, Tianjin Binhai, China [7]Big Data Technology Research Center, Nanhu Laboratory, Jiaxing, China [8]National Engineering Research Center For Software Engineering, Peking University, Beijing, China. Correspondence to: Junfeng Zhao, Yasha Wang <RihongQiu@stu.pku.edu.cn>.

*Proceedings of the $42^{nd}$ International Conference on Machine Learning*, Vancouver, Canada. PMLR 267, 2025. Copyright 2025 by the author(s).

## 1. Introduction

**Graph Neural Networks (GNNs)** (Cai et al., 2018; Kipf & Welling, 2017; Veličković et al., 2018; Zhang et al., 2025) have made significant advancements in recent years, establishing themselves as powerful tools for modeling complex graph structures across diverse domains. From social networks (Matsugu et al., 2023; Li et al., 2022c; Qin et al., 2022) to biochemistry (Duvenaud et al., 2015; Yang et al., 2023; Xu et al., 2023a), and extending to transportation systems (Gao et al., 2023a;b; Jiang et al., 2023b; Fang et al., 2023; 2024; Duan et al., 2024b) and financial networks (Li et al., 2022a; Huang et al., 2022; Liu et al., 2021; Duan et al., 2024a; Jiang et al., 2025), GNNs have proven their adaptability and effectiveness in a wide range of applications. Their exceptional versatility is evident in their ability to capture relational dependencies effectively, thereby facilitating data-driven decision-making processes.

Recently, **Large Graph Models (LGMs)** or Pretrained Graph Models (PLM) (Xia et al., 2024; Hu et al., 2020b; Li et al., 2024b; Ma et al., 2024; Zhang et al., 2023; Wang et al., 2024b; Jin et al., 2024; Jiang et al., 2024; Liu et al., 2023; Yu et al., 2024; 2025) have achieved breakthrough advancements by emulating the success of Large Language Models (LLMs) (OpenAI, 2023; Touvron et al., 2023), demonstrating powerful graph learning capabilities through large-scale graph data pretraining. Inspired by the paradigm of LLMs, LGMs such as GPT-GNN (Hu et al., 2020b), GPT-ST (Li et al., 2024b), and GPT-Het (Ma et al., 2024) have shown remarkable potential in capturing complex structural dependencies across diverse graph domains.

Despite the remarkable capabilities of LGMs in modeling complex graph relationships, they are hindered by significant limitations. One critical issue is that these models lack inherent **Emergent Abilities** (Zhang et al., 2023), which stem from the intricate nature of graph structures and the variability across different domains. As a result, LGMs require extensive and computationally intensive fine-tuning for downstream tasks through **Graph Continual Learning (GCL)**. This dependence on fine-tuning presents a crucial challenge for practical deployment: *How can we develop strategies to effectively distill task-specific training corpora*

*that enable efficient and effective fine-tuning of GNNs, ultimately enhancing their adaptability?*

Specifically, several challenges like **C1** and **C2** still remain unaddressed in the pursuit of the distilled small, synthetic, yet informative datasets, where GNNs trained on them can achieve competitive performance compared to those fine-tuned on large-scale graph datasets:

❶ *C1.* Downstream tasks frequently involve exceedingly large datasets with substantial noise, which results in ineffective fine-tuning and considerable computational overhead. The cost of using the whole large datasets in fine-tuning often rivals that of retraining an entire LGM, rendering such methods impractical for widespread applications.

❷ *C2.* The transfer of pretrained LGMs to different downstream tasks poses significant challenges due to structural mismatches. Specifically, current dataset distillation based GCL methods are limited to graph-level tasks (Xu et al., 2023b) and fail to generalize to tasks at the node level and edge level due to inconsistent input structures.

To address the aforementioned challenges, we introduce **Light**weight **G**raph **N**eural **T**angent **K**ernel (**LIGHTGNTK**), a novel dataset distillation framework tailored for effective GCL. Our method primarily focuses on distilling essential knowledge from large-scale graph datasets into compact, synthetic datasets via the Graph Neural Tangent Kernel (GNTK), which effectively captures both structural and feature relationships through the training gradients. To maintain the efficiency of conventional GNTK while minimizing computational overhead, LIGHTGNTK utilizes a low-rank approximation of the full Laplacian matrix derived via Bernoulli sampling on the eigenvalue spectrum. Theoretical guarantees support the quality of this approximation, and its computation is further accelerated by establishing a linear association with the sampled spectrum. Additionally, LIGHTGNTK incorporates a subgraph anchoring strategy to graph input structures across graph-, node-, and edge-level tasks, facilitating seamless adaptation to a variety of downstream applications In general, our contributions are summarized as follows.

- A unified dataset distillation framework for LGMs in GCL scenarios is proposed. By anchoring tasks to subgraph structures, LIGHTGNTK demonstrates adaptability to diverse downstream tasks, including node-level, edge-level, and graph-level applications.

- To address the computational complexity of GNTK, a Lightweight GNTK (LIGHTGNTK) is introduced. This framework balances structural and feature representation fidelity while minimizing dataset distillation overhead.

- Comprehensive experimental results demonstrate that LIGHTGNTK outperforms baseline methods across multi-

ple benchmark datasets and tasks, particularly in continual learning scenarios.

## 2. Related Work

### 2.1. Neural Tangent Kernel

The Neural Tangent Kernel (**NTK**) has emerged as a pivotal theoretical tool in understanding the training dynamics of neural networks (Jacot et al., 2020; Neal & Neal, 1996; Yang, 2019; Ren & Sutherland, 2025), with a wide range of application including neural architecture search (Park et al., 2020; Chen et al., 2022), meta-learning (Zhou et al., 2021), active learning (Holzmüller et al., 2023), and dataset distillation (Nguyen et al., 2021; Guo et al., 2024; Xu et al., 2023b). Originating from the concept of mapping infinitely wide neural networks to kernel methods, NTK simplifies the intricate training process of deep neural networks into kernel ridge regression (Jacot et al., 2020; Lee et al., 2020; Yang & Littwin, 2021).

In the realm of graph data, the Graph Neural Tangent Kernel (GNTK) extends the NTK framework to capture the relational and structural information inherent in graphs (Du et al., 2019; Xu et al., 2023b; Wang et al., 2024a). GNTK facilitates theoretical analysis and practical applications in graph-based learning tasks , and has been successfully applied to various graph-related tasks.

However, the practical adoption of NTK in large-scale learning tasks is hindered by its super-quadratic runtime complexity (Arora et al., 2019; Li et al., 2019; 2024a). To address this challenge, numerous studies have focused on accelerating NTK computations, such as using Monte Carlo method (Novak et al., 2018; Arora et al., 2019), low-rank approximations to kernel matrices (Alaoui & Mahoney, 2015; Avron et al., 2017; Zandieh et al., 2020), and Kronecker Sketching methods (Jiang et al., 2022). But these methods primarily target NTK in the context of standard neural networks and do not consider the unique characteristics of graphs. Building on these foundations, **our work introduces novel enhancements and unified definitions to improve computational efficiency and adapt GNTK to practical scenarios**, creating a more scalable and versatile framework for graph-based problems.

### 2.2. Dataset Distillation for Graph Continual Learning

Dataset distillation aims to synthesize a small but informative dataset to summarize the original large-scale datasets (Yu et al., 2023; Lei & Tao, 2023; Ding et al., 2024; Zhu et al., 2025), thereby expediting model training for continual learning, removing noise samples in the full datasets, and reducing computational overhead. Traditional methods like core set selection focus on identifying representative samples from original data (Goodman & O'Rourke, 1997),

which often fall short in capturing the complex high-order dependencies inherent in structured data. In contrast, recent advancements in meta-learning have introduced methods like gradient matching (Zhao et al., 2021) and trajectory matching (Miao et al., 2024; Zhu et al., 2025) in dataset distillation, which aligns distilled datasets and full datasets in the perspective of gradient or trajectory.

For graph structural data, emerging approaches focus on preserving structural and spectral properties while achieving efficient dataset distillation. Early methods address structural preservation by leveraging self-expressive models for graph reconstruction (Liu et al., 2024b) or maintaining consistent receptive field distributions to retain local neighborhood structures (Liu et al., 2022). Complementary to these, spectral-based approaches like GDEM (Liu et al., 2024a) align graph eigenbases to preserve spectral characteristics. In terms of optimization strategies, (Gupta et al., 2024) proposes a gradient-free distillation for node classification, while GCOND (Jin et al., 2022b) improves efficiency through gradient-based optimization.

A recently emerging direction leverages the Graph Neural Tangent Kernel (GNTK) (Guo et al., 2024; Xu et al., 2023b) to distill datasets by focusing on model dynamics learning. However, these methods encounter significant challenges, including low computational efficiency, insufficient utilization of graph structural information, and a lack of scalability to graph tasks at various levels.

# 3. Preliminaries

The Neural Tangent Kernel (NTK) describes the training dynamics of infinitely wide neural networks; however, its computation is generally intractable in practical applications. To address this limitation, the empirical Neural Tangent Kernel (eNTK) (Jacot et al., 2020) serves as a practical surrogate, facilitating empirical analyses of neural training dynamics. In Section 3.1, the definition of GNNs is first revisited. Then we review the computation of eNTK and its extension to Graph Neural Tangent Kernel (GNTK) for GNNs in Section 3.2. Additionally, we formally define the problem of graph dataset distillation in Section 3.3.

## 3.1. Revisiting Graph Neural Networks

We adopt the message-passing framework (Kipf & Welling, 2017) for graph neural networks, which is informed by spectral graph theory (Shuman et al., 2013). Consider a graph with an adjacency matrix $A \in \mathbb{R}^{n \times n}$ and node features $X \in \mathbb{R}^{n \times d}$ with node number $n$ and feature dimension $d$. The Laplacian matrix is defined as $L = \tilde{D}^{-1/2} \tilde{A} \tilde{D}^{-1/2}$ (Zhang et al., 2024; Li et al., 2022b), where $\tilde{A} = A + I$ and $\tilde{D}$ is the degree matrix. The Laplacian matrix $L$ can be diagonalized as $U \Lambda U^{\top}$, with $U$ representing the eigenvector matrix and

$\Lambda$ being a diagonal matrix of eigenvalues.

In the standard spatial (message-passing) formulation, the hidden representation at the $(l + 1)$-th layer, denoted as $H^{(l+1)}$, is computed as

$$H^{(l+1)} = \sigma\big(L\,H^{(l)}W^{(l)}\big), \qquad (1)$$

where $H^{(l)} \in \mathbb{R}^{n \times d_l}$ denotes the hidden representation at layer $l$, $H^{(0)} = X$, and $W^{(l)}$ are learnable parameters. $\sigma(\cdot)$ denotes an activation function (Jiang et al., 2023a; Kipf & Welling, 2017).

## 3.2. Revisiting eNTK & GNTK

**eNTK** The empirical Neural Tangent Kernel (eNTK), denoted by $\Theta_\theta(x_1, x_2)$, quantifies the similarity between two data points based on their gradient-space representation (Jacot et al., 2020; Mohamadi et al., 2022). Formally, for a neural network function $f_\theta(\cdot)$ parameterized by $\theta$, the eNTK is defined as follows:

$$\Theta_\theta(x_1, x_2) = \big[\nabla_\theta\big(f_\theta(x_1)\big)\big]\big[\nabla_\theta\big(f_\theta(x_2)\big)\big]^{\top}, \quad (2)$$

where $\nabla_\theta(f_\theta(x))$ represents the Jacobian of $f_\theta(x)$ with respect to the flattened parameter vector $\theta \in \mathbb{R}^P$, captures the rates of change of each output dimension of the function with respect to each input parameter. Specifically, if the model outputs $O$-dimensional vectors, then $\nabla_\theta(f_\theta(x)) \in \mathbb{R}^{O \times P}$, meaning that it has $O$ rows (one for each output dimension) and $P$ columns (one for each parameter). This structure leads to $\Theta_\theta(x_1, x_2) \in \mathbb{R}^{O \times O}$, reflecting the interactions between the gradients of the output dimensions for the two input data points. When considering two groups of data points of sizes $N_1$ and $N_2$, this results in $N_1 N_2$ such matrices. Stacking these $O \times O$ matrices yields a block-structured kernel of size $(N_1 O) \times (N_2 O)$, capturing pairwise gradient correlations across the dataset.

**GNTK** For graph-structured data, the Graph Neural Tangent Kernel (GNTK) extends this concept by analyzing the gradient of $f_\theta(\mathcal{G}) = f_\theta(L, X)$, where $f_\theta(\cdot)$ denotes a graph neural network (GNN). Here, $\mathcal{G}$ represents the graph sample, $L$ is the graph Laplacian, and $X$ is the node feature matrix. The GNTK between two graphs $\mathcal{G}_1$ and $\mathcal{G}_2$ is defined as:

$$\Theta_\theta(\mathcal{G}_1, \mathcal{G}_2) = [\nabla_\theta f_\theta(L_1, X_1)]\,[\nabla_\theta f_\theta(L_2, X_2)]^{\top}. \quad (3)$$

However, traditional spatial-domain methods for computing the GNTK involve evaluating gradients at the graph level, which necessitates computationally intensive operations that scale quadratically with the number of nodes $n$ (Shuman et al., 2013). Consequently, calculating the GNTK becomes prohibitive for large-scale graphs.

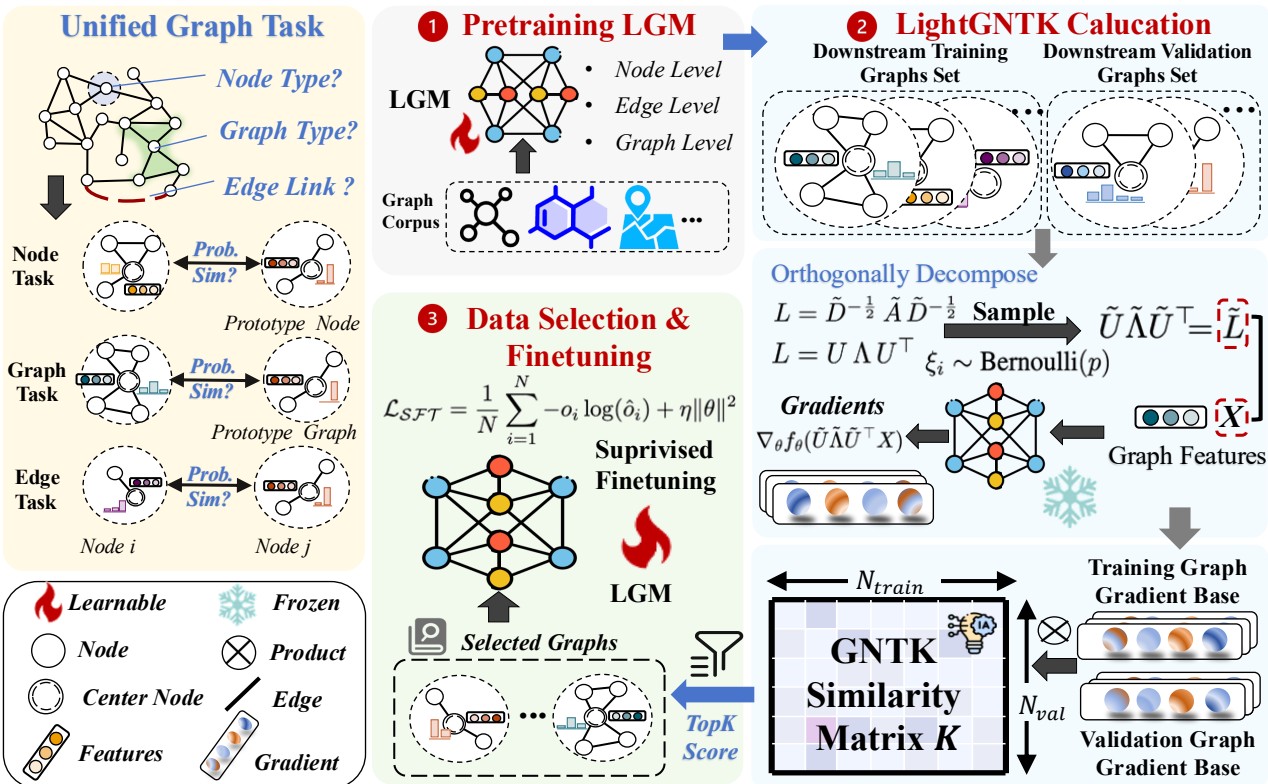

Figure 1. **The overall framework of LIGHTGNTK.** Based on the unified definition proposed in Section 4.1, we represent tasks at graph, node, and edge levels uniformly using subgraphs. ❶ Given a graph corpus, we pretrain the LGM in an unsupervised manner , utilizing data from node, edge, and graph levels. ❷ Next, we calculate the GNTK similarity matrix $K$ by measuring gradient dot products between training and validation graphs. To accelerate gradient computation, we employ a low-rank approximation of the graph Laplacian matrix $L$ via Bernoulli sampling. ❸ Finally, data selection is performed by identifying the $N_{syn}$ most informative samples according to the GNTK matrix. These selected samples are then used to fine-tune the LGM to improve its performance on downstream tasks.

## 3.3. Graph Dataset Distillation

Dataset distillation aims to condense a large dataset into a significantly smaller synthetic subset while preserving the essential information necessary for downstream tasks. In this work, we focus on addressing the problem of *graph dataset distillation* for classification tasks. Formally, given a target *training set* $\mathcal{D}_{train} = \{(\mathcal{G}_i, Y_i)\}_{i=1}^{N_{train}}$, our goal is to create a smaller *synthetic set* $\mathcal{D}_{syn} = \{(\tilde{\mathcal{G}}_i, \tilde{Y}_i)\}_{i=1}^{N_{syn}}$ where $N_{syn} \ll N_{train}$, such that a GNN trained on $\mathcal{D}_{syn}$ maintains performance comparable to one trained on $\mathcal{D}_{train}$. Concretely, the distillation process proceeds as follows: (1) A *distillation procedure* generates $\mathcal{D}_{syn}$ from $\mathcal{D}_{train}$. (2) A *downstream classifier* is trained on $\mathcal{D}_{syn}$ and evaluated on a test set $\mathcal{D}_{test}$, which is disjoint from both $\mathcal{D}_{train}$ and $\mathcal{D}_{syn}$. The evaluation on $\mathcal{D}_{test}$ reflects how effectively $\mathcal{D}_{syn}$ retains the knowledge contained in $\mathcal{D}_{train}$.

## 4. Method

In this section, we introduce LIGHTGNTK, a novel framework designed to efficiently and effectively distill graph datasets for Graph Continual Learning (GCL), as illustrated in Figure 1. ❶ First, as described in Section 4.1, we propose a unified task definition that integrates tasks at the node, graph, and edge levels within a subgraph-centric framework. This approach facilitates the extraction of task-specific subsets from large-scale training data, ensuring that these samples are well-suited for a variety of downstream tasks. ❷ Next, in Section 4.2, we provide a theoretical analysis of the GNTK and introduce an efficient spectral-domain approximation, LIGHTGNTK, which significantly reduces both time and space complexity compared to standard GNTK evaluations. We also demonstrate that our method yields an approximation to the GNTK with a tight approximation error upper bound. ❸ Finally, in Section 4.3.2, we describe the pretraining and fine-tuning paradigm of the LGM through an unsupervised learning task. For improved readability, the

main notations are summarized in Table 4 of Appendix A.

### 4.1. Unified Task Definition

Let graph sample set $\mathbb{G} = \{\mathcal{G}_1, \ldots, \mathcal{G}_m\}$ be a set of graphs. Each graph $\mathcal{G}_i = (V_i, E_i, X_i, A_i, Y_i)$ is defined as follows: $V_i$ is the set of nodes, with $|V_i| = n_i$; $E_i \subseteq V_i \times V_i$ is the set of edges; $X_i \in \mathbb{R}^{n_i \times d}$ denotes the node feature matrix; $A_i \in \mathbb{R}^{n_i \times n_i}$ is the adjacency matrix; $Y_i$ denotes the labels associated with nodes, edges, or the entire graph.

In this paper, we propose a subgraph-centric framework for the unified handling of tasks at the node, edge, and graph levels. Given a target (node, edge, or graph), we construct a *query subgraph* $\mathcal{G}^{\mathcal{Q}}$ by including the target and its neighbors within $k$ hops.[1] We then compare this query subgraph to *prototype subgraphs* from a support set $\mathcal{D}_{\text{syn}}$ using a similarity function $\text{sim}(\cdot, \cdot)$. ❶ **For Node-Level and Graph-Level Tasks.** Let $C$ denote the set of class labels, and assume $y_i \in C$ is the label for a node or a graph instance. After extracting the query subgraph $\mathcal{G}^{\mathcal{Q}}$ for the target node or graph, we compute the output logits $o_i$ for the query. The predicted label is given by $y_i = \arg\max_{c \in C} \text{sim}(o_i, \tilde{o}_c)$, where the prototype logits $\tilde{o}_c$ for class $c$ can be selected as the average logits of subgraph instances in the support set $\mathcal{D}_{\text{syn}_c}$ for class $c$: $\tilde{o}_c = \frac{1}{|\mathcal{D}_c|} \sum_{(\mathcal{G}_j, y_j) \in \mathcal{D}_c} o_j$, or as a one-hot vector for class $c$, i.e., $\tilde{o}_c[c] = 1$ and zeros elsewhere. ❷ **For Edge-Level Tasks.** Given a candidate pair $(v_i, v_q)$ for edge prediction, subgraphs $\mathcal{G}_i^{\mathcal{Q}}$ and $\mathcal{G}_q^{\mathcal{Q}}$ centered at $v_i$ and $v_q$ are constructed, yielding logits $o_i$ and $o_q$. An edge is predicted between $v_i$ and $v_q$ if there exists any neighbor $v_j$ of node $v_i$ such that $\text{sim}(o_i, o_q) \geq \max_{v_j \neq v_q}(\text{sim}(o_i, o_j)) + \epsilon$, where $\epsilon \geq 0$ is a margin hyperparameter. Notably, this criterion can also be applied symmetrically by considering the neighborhood of $v_q$ in place of $v_i$.

### 4.2. LIGHTGNTK: A Lightweight Approximation of Graph Neural Tangent Kernel

#### 4.2.1. LIGHTGNTK DEFINITION

To address the considerable computational overhead associated with conventional GNTK, we introduce a series of techniques aimed at significantly enhancing its computational efficiency.

❶ **Low-Rank Optimization** Building on critical insights from spectral graph theory (Von Luxburg, 2007), we tackle the high computational cost of GNTK by proposing a low-rank approximation of the graph Laplacian matrix $L$. In the spectral domain, each eigenvalue of the Laplacian encapsulates unique topological information. Importantly, we observe that both theoretical and empirical observa-

tions indicate that retaining only **a subset of eigenvalues and their associated eigenvectors is sufficient to accurately approximate GNN gradients** (Bianchi et al., 2020; Deng et al., 2022), thereby eliminating the need for full-rank Laplacian matrices in GNTK computation. To achieve this low-rank approximation, we perform Bernoulli sampling on the eigenvalues obtained from the spectral decomposition of Laplacian matrix, effectively balancing high-frequency and low-frequency components while preserving both local and global graph structures. Formally, given a graph Laplacian $L = U\Lambda U^\top$, where the eigenvalue matrix is $\Lambda = \text{diag}(\lambda_1, \ldots, \lambda_n)$ and the eigenvector matrix is $U = [u_1, u_2, \ldots, u_n]$, we define the expected rank $r$ and sample probability $p = \frac{r}{n}$. For each eigenvalue $\lambda_i$, we sample $\xi_i \sim \text{Bernoulli}(p)$ and form $\tilde{\Lambda} = \text{diag}(\xi_1\lambda_1, \ldots, \xi_n\lambda_n) \in \mathbb{R}^{r \times r}, \xi_i = 1$. Eigenvectors in $\tilde{U} \in \mathbb{R}^{n \times r}$ with nonzero $\xi_i$ are retained accordingly.

Next, to compute $\tilde{L}X = \tilde{U}\tilde{\Lambda}\tilde{U}^\top X$ efficiently, we decompose the computation into three sequential steps: ❶ $Z = \tilde{U}^\top X$, where $Z \in \mathbb{R}^{r \times d}$; cost: $O(nrd)$; ❷ $Y = \tilde{\Lambda}(Z^\top)$, where $Y \in \mathbb{R}^{r \times d}$; cost: $O(r^2 d)$; ❸ $\tilde{L}X = \tilde{U}Y$; cost: $O(nrd)$. Thus, the overall computational complexity is $O(nrd) + O(r^2 d) + O(nrd) \approx O(nrd)$, where $r \ll n$. This significantly reduces the complexity compared to the naive approach of $O(n^2 d)$, which is conceptually related to linear attention (Katharopoulos et al., 2020). The resulting low-rank Laplacian $\tilde{L} = \tilde{U}\tilde{\Lambda}\tilde{U}^\top$ preserves essential structural information, and the low-rank approximation of GNTK is given by:

$$\Theta(\mathcal{G}_i, \mathcal{G}_j) = \left[\nabla_\theta f_\theta(\tilde{U}_i \tilde{\Lambda}_i \tilde{U}_i^\top X_i)\right] \cdot \left[\nabla_\theta f_\theta(\tilde{U}_j \tilde{\Lambda}_j \tilde{U}_j^\top X_j)\right]^\top . \tag{4}$$

❷ **Structural Optimizations** Furthermore, we implement three additional acceleration strategies: (1) For node-level and edge-level tasks, instead of performing eigenvalue decomposition of the Laplacian matrix $L$ for the entire graph, we conduct the decomposition within subgraphs, thereby reducing computational complexity due to the smaller matrix sizes. (2) For extremely large graph-level tasks, we adopt a batch acceleration technique based on Monte Carlo sampling (see Appendix E) to enhance efficiency. (3) When collecting gradients, we utilize only the gradients from the final layer rather than from all layers, achieving a balance between speed and accuracy.

#### 4.2.2. APPROXIMATION QUALITY OF LIGHTGNTK

We now formalize the relationship between the gradients obtained with the Bernoulli-sampled Laplacian $\tilde{L}$ and those obtained with the full-rank Laplacian $L$. Throughout, we assume that the GNN $f_\theta(\cdot)$ is $L_\nabla$-smooth with respect to its input as defined in (Heinonen, 2005).

---

[1]In graph-level tasks, the entire graph is treated as one subgraph, and a virtual node is attached to all nodes.

**Theorem 4.1** (Gradient Approximation Error). *Let $X \in \mathbb{R}^{n \times d}$ be the node-feature matrix and let $L = U\Lambda U^\top$ be the graph Laplacian with eigen-pairs $\{(\lambda_i, u_i)\}_{i=1}^n$. Construct $\tilde{L}$ via independent Bernoulli sampling with probability $p$ on every eigenvalue, i.e. $\tilde{L} = \tilde{U}\tilde{\Lambda}\tilde{U}^\top$ with $\tilde{\Lambda}_{ii} = \xi_i \lambda_i$, $\xi_i \sim$ Bernoulli$(p)$. Then the expect of gradient approximation error is:*

$$
\begin{aligned}
\mathbb{E}_\xi &\left[\left\|\nabla_\theta f_\theta(\tilde{L}X) - \nabla_\theta f_\theta(LX)\right\|_F^2\right] \\
&\leq L_\nabla^2 (1-p) \sum_{i=1}^n \lambda_i^2 \left\|u_i^\top X\right\|_F^2,
\end{aligned}
\tag{5}
$$

*where $L_\nabla$ is the Lipschitz smoothness coefficient. The full proof can be found in Appendix F.2.*

**Theorem 4.2** (LIGHTGNTK Approximation Error Upper Bound). *Let $\mathcal{G}_1$ and $\mathcal{G}_2$ denote two graphs with corresponding node feature matrices $X_1$ and $X_2$. The approximation error of GNTK computed using the Bernoulli-sampled Laplacian $\tilde{L}$ can be bounded from below as follows:*

$$
\left|\Theta_{\text{LIGHTGNTK}}(\mathcal{G}_1, \mathcal{G}_2) - \Theta_{GNTK}(\mathcal{G}_1, \mathcal{G}_2)\right| \leq L_\nabla \left(\mathcal{C}_1 \Delta_2 + \mathcal{C}_2 \Delta_1\right),
$$

*where $\Delta = \sqrt{(1-p) \cdot \sum_{i=1}^n \lambda_i^2 \left\|u_i^\top X\right\|^2}$ denotes the upper bound on the approximation error, as derived from the gradient error analysis. Here, $\mathcal{C} = \|\nabla_\theta f_\theta(LX)\|$ is a constant that depends only on the network $\theta$ and graph data $\mathcal{G}$. Therefore, as the sampling probability $p$ increases, the approximation error of GNTK decreases. The complete proof is provided in Appendix F.3.*

**Time Complexity Analysis** Compared with the conventional GNTK, which requires $O(n^2 d)$ time due to full-rank Laplacian operations, our low-rank optimization reduces the time complexity to $O(nrd)$, where $r \ll n$ is the chosen rank. This substantial reduction is achieved by only retaining $r$ spectral components, allowing all Laplacian-related computations to be executed efficiently with much smaller matrices. As a result, our approach scales to larger graphs and offers significant computational savings over traditional GNTK methods without sacrificing accuracy.

## 4.3. Graph Dataset Distillation Instantiation

This section first presents a theoretical analysis of the relationship between training and test samples, which forms the basis for the proposed graph data sampling criteria. Then, the corresponding graph distillation and model training pipeline is described in detail.

### 4.3.1. THEORETICAL ANALYSIS OF THE RELATIONSHIP BETWEEN TRAIN & TEST DATA

We theoretically analyze the influence of training samples gradients $\mathcal{G}_o = \{L_o, X_o\}$ on test samples $\mathcal{G}_u = \{L_u, X_u\}$

through the lens of the GNTK, as detailed in Appendix C. The analysis demonstrates that the change in test sample logits induced by a training step on $\mathcal{G}_o$ is proportional to the GNTK value $\Theta(\mathcal{G}_u, \mathcal{G}_o)$ between the training and test samples, quantitatively capturing the transfer of gradient information across the train-test split (Guo et al., 2024). Consequently, filtering or reweighting training samples based on GNTK values with respect to the test set can effectively reduce undue train-test coupling and improve the generalization performance of dataset distillation methods (Guo et al., 2024).

### 4.3.2. INSTANTIATION PIPELINE

Our practical pipeline consists of three main steps. First, we pretrain the LGM using an unsupervised objective to obtain robust graph representations. Next, for each candidate graph $\mathcal{G}_c$ in the training set, we evaluate its quality according to the framework of LIGHTGNTK, and select the top $N_{\text{syn}}$ query subgraphs to construct the synthetic set $\mathcal{D}_{\text{syn}}$. Finally, we use $\mathcal{D}_{\text{syn}}$ to train the downstream classifier and evaluate its predictive performance.

❶ **Pretrain LGM** We pretrain LGM in an unsupervised fashion to jointly learn node representations and graph structural patterns. Following the pretraining paradigm of masked signal reconstruction and link prediction (as in GPT-GNN (Hu et al., 2020b)), we randomly mask a subset of node features and edges, and train the model to reconstruct the masked information on large-scale unlabeled graphs. This allows LGM to learn more expressive representations for nodes and structures, which facilitates effective transfer to various downstream graph tasks.

❷ **LIGHTGNTK Calculation** The theoretical analysis in Section 4.2 demonstrates that performance on downstream tasks can be improved by fine-tuning LGM using training samples that are similar to those from the test set. In practice, however, test data are not available during training process. Under the assumption that the validation and test sets are independently drawn from the same distribution, we use the validation set to guide the selection of relevant training samples. Specifically, we make the following assumption.

***Assumption. Validation-Test Distribution Match*** *In real-world deployment scenarios, we assume that both the validation set $\mathcal{D}_{val}$ and the test set $\mathcal{D}_{test}$ are independently sampled from the same target distribution $\Pi_{target}$, such that:*

$$
\Pi_{\mathcal{D}_{\text{val}}} \equiv \Pi_{\mathcal{D}_{\text{test}}} \sim \Pi_{\text{target}}, \tag{6}
$$

*where $\Pi_{\mathcal{D}_.}$ denotes the underlying data distribution from which dataset $\mathcal{D}_.$ is sampled.*

This alignment is a standard practice in machine learning evaluation and deployment (Recht et al., 2019; Wang et al.,

2022; Lohr, 2021; Jiang & Wang, 2023), which ensures that validation metrics reliably predict test performance during dataset distillation. To further substantiate the distribution match assumption, we conduct quantitative analyses in Appendix D.4. These results provide strong empirical evidence that the validation and test sets are statistically similar.

Based on the assumption, we leverage the validation set as a proxy for training sample selection. To measure the similarity between the gradients of training and validation samples, we utilize the GNTK, as introduced in Section 4.2. Specifically, we compute a cross-set kernel matrix $K \in \mathbb{R}^{N_{\text{train}} \times N_{\text{val}}}$, where $N_{\text{train}}$ and $N_{\text{val}}$ denote the sizes of the training and validation sets, respectively. Each entry $K_{ij} = \Theta_\theta(\mathcal{G}_1, \mathcal{G}_2)$ quantifies the gradient covariance between training graph $\mathcal{G}_i$ and validation graph $\mathcal{G}_j$ under GNN parameters $\theta$. Furthermore, motivated by the theoretical analysis in Section 4.2, we adopt a computationally efficient variant of GNTK by extracting gradients from a specific layer and computing a lightweight similarity measure.

**❸ Data Selection & Supervised Finetuning**   We define the LIGHTGNTK-score $\mathcal{I}(\mathcal{G}_c)$ for each candidate graph $\mathcal{G}_c \in \mathcal{D}_{\text{train}}$ as the minimum gradient similarity between $\mathcal{G}_c$ and all graphs in the validation set, that is,

$$\mathcal{I}(\mathcal{G}_c) = \min_{\mathcal{G}_v \in \mathcal{D}_{\text{val}}} \Theta(\mathcal{G}_c, \mathcal{G}_v). \tag{7}$$

Based on the computed LIGHTGNTK-scores, we construct the synthetic set $\mathcal{D}_{\text{syn}}$ by selecting those candidate graphs with the lowest LIGHTGNTK scores, as defined in Eq. (8):

$$\mathcal{D}_{\text{syn}} = \{\mathcal{G}_c \in \mathcal{D}_{\text{train}} \mid \mathcal{I}(\mathcal{G}_c) \in \text{Min-}N_{\text{syn}}(\{\mathcal{I}(\mathcal{G}_i)\}_{i=1}^{N_{\text{train}}})\}, \tag{8}$$

where Min-$N_{\text{syn}}(\cdot)$ indicates selecting the $N_{\text{syn}}$ lowest elements from the given set.

Next, we utilize the selected samples to perform supervised fine-tuning of the pretrained LGM. Let $z$ denote the output from the last layer of the LGM. The predicted class probabilities are computed via the softmax function: $\hat{y} = \text{Softmax}(z)$, where $\hat{o}_k = \exp(o_k) / \sum_{j=1}^{|O|} \exp(z_j)$ for class $k$. To support different prediction tasks (edge, node, and graph levels) centered on a subgraph's node, we adopt a unified loss function:

$$\mathcal{L}_{\text{SFT}} = \frac{1}{N} \sum_{i=1}^{N} -o_i \log(\hat{o}_i) + \eta \|\theta\|^2, \tag{9}$$

where $N$ is the number of training samples, $|O|$ is the number of output classes, $\hat{o}_i$ represents the logits assigned by the model to the correct class for sample $i$, and $o_i$ denotes the ground-truth logits for class $i$. The second term is an $L_2$ penalty (weight decay) applied to the learnable parameters $\theta$ of the GNN, scaled by a hyperparameter $\eta$ to prevent overfitting.

## 5. Experiment

To evaluate the effectiveness of LIGHTGNTK in graph dataset distillation, we conduct extensive experiments on 13 datasets, including tasks at node, edge, and graph levels, and compare our method with state-of-the-art baselines. The experiments aim to address the following research questions:

- **RQ1:** How does LIGHTGNTK compare with state-of-the-art methods in graph dataset distillation for GCL tasks at node, edge, and graph levels?

- **RQ2:** Does LIGHTGNTK achieve improved efficiency compared to traditional GNTK?

- **RQ3:** What are the contributions of the key components towards improving LIGHTGNTK's overall performance?

Further details and experimental results are provided in Appendix D.

### 5.1. Experimental Setup

**Datasets.**   We conduct a comprehensive evaluation using a total of 13 datasets across three types of graph learning tasks. For graph classification , we utilize 7 datasets: *NCI1*, *NCI109*, *PROTEINS*, and *DD* from TUDataset (Morris et al., 2020), as well as *ogbg-molhiv*, *ogbg-molbbbp*, and *ogbg-molbace* from Open Graph Benchmarks (OGB) (Hu et al., 2020a). For node classification , we employ 4 citation network datasets: *Cora*, *CiteSeer*, and *PubMed* from the Planetoid Dataset (Yang et al., 2016), along with *ogbn-arxiv* from OGB. For link prediction , we include 2 datasets: *ogbl-collab* and *ogbl-ddi*, both from OGB. Further details for all datasets are provided in Table D.1 of Appendix D.1.

**Methods and Baselines.**   Our experimental framework compares LIGHTGNTK  with the SOTA dataset distillation methods for graph. We evaluate two versions of our proposed framework: (i) GNTK, which directly leverages the Laplacian matrix for feature selection without any optimization; and (ii) the lightweight variant, LIGHTGNTK, which incorporates efficiency optimizations. To ensure a comprehensive evaluation, we extend our baseline analysis to encompass three distinct categories: ❶ *Coreset selection methods*, including Random Sampling, Herding (Welling, 2009), and K-Center Greedy Selection (Farahani & Hekmatfar, 2009; Sener & Savarese, 2017); ❷ *Learning-based graph distillation*, represented by DosCond (Jin et al., 2022a); ❸ *Kernel approximation techniques*, exemplified by KIDD (Xu et al., 2023b), which uses ridge regression for graph-level tasks to approximate GNTK computations. Detailed implementation specifics and theoretical foundations of all baseline methods are systematically documented in Appendix D.3.

*Table 1.* Performance comparison (mean±std) on graph classification tasks. The best and second-best results are **bolded** and underlined.

| Name | Graphs/Cl | Ratio | Random | Herding | K-Center | DosCond | KIDD | GNTK | LIGHTGNTK | Whole Dataset |
|---|---|---|---|---|---|---|---|---|---|---|
| NCI1 (ACC) | 1 | 0.06% | 57.4±3.0 | 59.2±3.0 | 59.2±3.0 | 57.1±0.9 | 60.1±0.9 | 59.4±0.4 | **62.0±7.1** | |
| | 10 | 0.61% | 59.9±2.0 | 62.8±0.9 | 59.1±0.8 | 60.8±0.9 | 61.7±1.3 | **64.8±0.2** | 64.4±3.3 | 80.0±1.1 |
| | 50 | 3.04% | 60.5±2.1 | 62.5±2.0 | 59.5±0.5 | 62.7±0.8 | 64.2±0.6 | 64.5±0.4 | **66.4±4.0** | |
| NCI109 (ACC) | 1 | 0.06% | 54.3±2.3 | 51.7±0.9 | 51.7±0.9 | 54.9±2.3 | 55.2±2.7 | **57.5±0.3** | 55.3±3.7 | |
| | 10 | 0.61% | 61.9±1.6 | 63.6±0.3 | 52.9±1.7 | 61.4±1.5 | 63.5±0.5 | **64.9±0.1** | 64.2±4.3 | 77.7±0.6 |
| | 50 | 3.03% | 64.0±1.4 | 64.7±1.2 | 55.0±2.1 | 62.9±1.6 | **70.4±1.3** | 66.8±2.7 | 65.6±2.5 | |
| PROTEINS (ACC) | 1 | 0.22% | 57.8±1.8 | 67.6±1.7 | 67.6±1.7 | 63.4±1.9 | 68.8±3.9 | **69.3±6.4** | 68.1±1.2 | |
| | 10 | 2.25% | 67.2±0.7 | 68.3±1.0 | 71.4±3.3 | 71.7±0.4 | 74.1±1.9 | 73.6±0.8 | **74.7±5.5** | 78.6±2.6 |
| | 50 | 11.24% | 69.6±4.0 | 70.1±1.0 | 72.9±2.6 | 73.2±0.8 | 75.0±1.9 | 74.0±3.9 | **75.9±3.2** | |
| DD (ACC) | 1 | 0.21% | 61.3±8.5 | 60.7±8.4 | 61.0±3.2 | 63.0±0.7 | 65.8±1.7 | **68.6±0.4** | 65.8±5.4 | |
| | 10 | 2.12% | 66.8±2.1 | 67.4±0.7 | 66.2±2.4 | 68.1±1.8 | 70.6±1.3 | 71.4±0.1 | **71.5±1.7** | 76.9±3.2 |
| | 50 | 10.62% | 71.4±2.2 | 71.6±1.9 | 72.3±1.0 | 70.9±1.0 | 73.1±2.2 | **73.3±0.4** | 72.9±6.5 | |
| ogbg-molhiv (ROC-AUC) | 1 | 0.01% | 55.5±3.6 | 63.3±2.5 | 63.3±2.5 | 61.2±2.5 | 63.3±1.6 | **64.8±0.5** | 64.1±7.2 | |
| | 10 | 0.06% | 57.9±1.2 | 62.1±0.9 | 63.0±1.3 | 64.7±3.1 | 67.5±5.4 | **69.2±0.2** | 68.5±8.7 | 75.0±0.7 |
| | 50 | 0.30% | 62.3±1.4 | 61.6±1.4 | 62.8±1.2 | 62.0±1.8 | **70.8±6.2** | 67.4±0.2 | 69.3±8.3 | |
| ogbg-molbbbp (ROC-AUC) | 1 | 0.12% | 57.9±2.0 | 62.8±1.2 | 62.8±1.1 | 58.4±3.0 | 62.8±1.1 | **64.4±0.1** | 64.0±4.2 | |
| | 10 | 1.23% | 55.6±0.0 | 62.5±0.0 | 59.6±1.6 | 62.1±1.3 | 64.4±1.5 | 63.8±0.1 | **64.6±6.1** | 65.0±1.4 |
| | 50 | 6.13% | 61.0±0.7 | 63.0±0.7 | 59.5±1.5 | 62.8±1.2 | 66.2±3.0 | 64.2±0.1 | **68.2±0.6** | |
| ogbg-molbace (ROC-AUC) | 1 | 0.17% | 63.8±0.9 | 54.6±3.8 | 54.6±3.8 | 66.7±2.1 | 69.3±1.6 | 68.2±0.2 | **69.4±7.1** | |
| | 10 | 1.65% | 64.9±1.7 | 56.1±4.1 | 65.8±5.8 | 69.4±1.4 | **74.8±2.0** | 71.2±0.5 | 70.8±9.6 | 72.7±1.7 |
| | 50 | 8.26% | 65.5±2.0 | 70.3±1.2 | 66.2±1.3 | 71.0±0.6 | 76.6±1.2 | 76.6±0.5 | **76.8±6.2** | |

**Settings and Evaluation.** For the graph classification tasks in TUDataset, we split each dataset into 80%, 10%, and 10% for training, validation, and test sets, respectively, and use accuracy (ACC) as the evaluation metric. For the Planetoid and OGB datasets, we follow the official splits and corresponding evaluation metrics. The detailed evaluation metrics used in the experiments are provided in in Table 6 at Appendix D.2. Throughout all experiments, we use Graph Isomorphism Network (GIN) (Xu et al., 2019) as the GNN backbone for our LGMs. To comprehensively assess the effectiveness of various distillation methods, We assess classification performance at the graph, node, and edge levels by distilling datasets in which each class contains 1, 10, or 50 samples at the corresponding level.

**Reproducibility.** To ensure reproducibility, we optimize the parameters of baseline models using the Adam Optimizer with $L_2$ regularization. During the pretrain process of the GNN backbone, a binary classification task is employed to distinguish between positive and negative edge samples on graphs. For the downstream tasks, a classification head is added above GNN backbone to predict. Furthermore, An early-stopping strategy with a patience of 50 epochs is employed to mitigate overfitting.

## 5.2. Dataset Distillation Results (RQ1)

The evaluation results for the graph-level and node-level tasks are summarized in Table 1 and Table 2 respectively, while edge-level results are reported in Table 8 in Appendix D.5. Based on these experiments, we identify the following key findings:

**Outperforming Existing Methods.** The proposed dataset distillation method LIGHTGNTK demonstrates superior performance compared to almost all baseline methods across various tasks, including graph, node, and edge levels. Notably, LIGHTGNTK can surpass baselines on most datasets, highlighting its ability to reduce data dependency while maintaining high predictive accuracy. These results clearly demonstrate that our method provides a more effective and reliable solution for efficient graph dataset distillation.

**Effective Learning from Limited Data** We observe that the model under dataset distillation method LIGHTGNTK achieves excellent performance across all three levels, even when trained with as few as one sample per class. This highlights the effectiveness and representativeness of the selected training samples, which are capable of capturing the essential characteristics of each class. Remarkably, even with less than 10% of the original training data, the model demonstrates performance that is nearly comparable to the results obtained when using the entire dataset. This suggests that LIGHTGNTK is highly efficient in sampling and leveraging the available data, and its ability to generalize effectively with limited information underscores the robustness of the distillation method.

## 5.3. Computational Efficiency Analysis (RQ2)

In addition to evaluating predictive accuracy, it is crucial to assess the computational efficiency of our approach. LIGHTGNTK is designed to significantly improve efficiency by leveraging a theoretically principled approximation of GNTK. We report the computation time of GNTK and

*Table 2.* Performance comparison (mean±std) on node classification tasks. The best and second-best results are **bolded** and underlined.

| Name | Nodes/Cl | Ratio | Random | Herding | K-Center | DosCond | GNTK | LIGHTGNTK | Whole Dataset |
|------|----------|-------|--------|---------|----------|---------|------|-----------|---------------|
| CiteSeer | 1 | 0.18% | 38.3±6.8 | 34.4±5.9 | 36.8±4.4 | 42.2±5.7 | 43.8±2.1 | **44.7±2.4** | |
| | 10 | 1.80% | 56.7±2.7 | 63.7±0.9 | 60.3±1.2 | 61.9±0.8 | 62.3±0.4 | **62.7±5.1** | 63.6±1.3 |
| | 50 | 9.02% | 64.0±2.6 | 63.7±1.2 | 64.6±3.2 | 63.5±1.1 | **64.8±1.6** | 63.4±1.1 | |
| Cora | 1 | 0.26% | 35.9±9.0 | 37.3±4.5 | 39.4±6.2 | 39.3±2.4 | 42.7±5.6 | **44.6±7.2** | |
| | 10 | 2.58% | 74.0±2.2 | 75.7±1.8 | 72.8±2.6 | 74.2±2.6 | 75.5±3.9 | **76.2±0.6** | 77.6±1.0 |
| | 50 | 12.92% | 77.6±1.0 | 77.4±1.0 | 77.7±1.1 | 77.4±0.9 | 77.6±1.0 | **77.9±2.5** | |
| PubMed | 1 | 0.02% | 53.8±5.0 | 52.9±6.1 | 52.9±6.2 | 55.6±2.7 | **56.4±3.0** | 55.8±1.8 | |
| | 10 | 0.15% | 65.8±3.2 | 65.5±4.1 | 64.5±3.9 | 61.0±3.8 | 70.1±2.0 | **70.7±2.1** | 69.7±2.9 |
| | 50 | 0.76% | 71.3±4.1 | 71.8±1.4 | 70.3±0.7 | 69.8±4.1 | **72.3±0.8** | 71.2±3.1 | |
| ogbn-arxiv | 1 | 0.02% | 11.5±6.6 | 14.5±3.0 | 14.5±3.0 | 13.2±0.8 | **15.7±1.4** | 15.3±0.7 | |
| | 10 | 0.24% | 16.1±2.6 | 15.3±1.7 | 16.4±2.1 | 16.3±4.5 | 18.2±0.8 | **18.9±3.2** | 51.4±1.4 |
| | 50 | 1.18% | 25.1±2.1 | 24.4±1.3 | 22.0±1.4 | 24.0±1.8 | 28.5±1.4 | **29.1±0.4** | |

LIGHTGNTK during the distillation process across all graph-level datasets, as summarized in Table 3. As demonstrated in the table, the computation time required by LIGHT-GNTK is consistently lower than that of the original GNTK across all benchmarks, with reductions ranging from 6% to 27% depending on the graph size of dataset.

These results demonstrate that our approximation provides substantial efficiency gains. Importantly, as shown in previous sections, this improvement is achieved without sacrificing the representational power or predictive performance of GNTK. This improved efficiency enables the practical scaling of kernel-based graph methods to large real-world datasets.

*Table 3.* Computation time (in seconds) of GNTK and LIGHT-GNTK on graph classification datasets.

| Dataset | GNTK | LIGHTGNTK |
|---------|------|-----------|
| NCI1 | 196 | 144 |
| NCI109 | 184 | 136 |
| PROTEINS | 26.3 | 24.8 |
| DD | 278 | 203 |
| ogbg-molhiv | 921 | 683 |
| ogbg-molbbbp | 72.6 | 59.7 |
| ogbg-molbace | 54.5 | 48.8 |

**5.4. Key Components Study (RQ3)**

To demonstrate the effectiveness of key components in LIGHTGNTK, we conduct the following explorations. ❶ We explore different low-rank optimization strategies, including selecting the $r$ largest eigenvalues (top-$r$ sampling), selecting the $r$ smallest eigenvalues (bottom-$r$ sampling), and selecting $r$ eigenvalues based on Bernoulli sampling, where $r$ is the rank of the approximated Laplacian matrix. The comparative performance of these methods is summarized in Table 9 in Appendix D.6. ❷ We evaluated the quality and

computational cost of our distillation method under different sampling rates $p$ in Bernoulli sampling, and represent the result in Table 10 and Table 11 in Appendix D.7.

These experimentals demonstrate the effectiveness of our eigenvalue sampling strategy. Among various sampling strategies, Bernoulli sampling achieves the best performance by capturing both global and local eigenvalue information on the entire graph. Moreover, as the sampling rate increases, computational overhead increases rapidly, but performance improvements levels off after a certain threshold. Based on empirical observations, we set the sampling rate to 0.1 in experiments, which offers a favorable trade-off between performance and computational efficiency.

# 6. Conclusion and Future Work

In this paper, we propose Lightweight Graph Neural Tangent Kernel (LIGHTGNTK), a novel framework for efficient dataset distillation in Graph Continual Learning (GCL). LIGHTGNTK leverages a low-rank Laplacian approximation via Bernoulli sampling and linear associations in GNTK, substantially reducing computation while maintaining strong performance on graph, node, and edge tasks through a unified subgraph anchoring strategy. Extensive experiments on diverse benchmarks show that LIGHT-GNTK achieves state-of-the-art performance and provides a scalable, cost-effective solution for continual learning.

In future work, we will investigate the effects of distributional shifts between pretrain and fine-tuning datasets and explore methods to further improve model generalization and transferability in large-scale dynamic graphs.

# Acknowledgments

This work is supported by the National Natural Science Foundation of China (No.U23A20468).

## Impact Statement

This paper advances the efficiency and adaptability of Graph Neural Networks through a lightweight dataset distillation framework for large graph models. We anticipate positive societal impact by lowering the computational barriers to graph-based machine learning across diverse domains. We do not foresee any ethical issues or negative societal consequences beyond those generally associated with the advancement of machine learning.

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

# A. Notations

The main notations in this paper are summarized in Table 4.

*Table 4.* Notations Tables in LIGHTGNTK

| Notation | Definition |
|---|---|
| $\mathcal{G}$ | graph |
| $X$ | node feature matrix |
| $A$ | adjacency matrix |
| $Y$ | label matrix |
| $L$ | graph Laplacian matrix |
| $\Lambda$ | eigenvalues matrix of $L$ |
| $D$ | degree matrix of graph |
| $U$ | eigenvectors matrix of $L$ |
| $H$ | hidden representation in GNN |
| $n$ | number of nodes in a graph |
| $d$ | dimension of node features |
| $k$ | $k$ hop for subgraph |
| $r$ | expected rank of $L$ |
| $\Theta$ | Neural Tangent Kernel |
| $K$ | Neural Tangent Kernel Matrix |
| $f$ | neural network function |
| $\theta$ | parameters of neural network |
| $\mathcal{D}_{\text{train}}$ | target training subgraphs set |
| $\mathcal{D}_{\text{val}}$ | validation subgraphs set |
| $\mathcal{D}_{\text{syn}}$ | synthetic training subgraphs set |
| $\mathcal{D}_{\text{test}}$ | test subgraphs set |
| $N_{\text{train}}$ | number of subgraphs in $\mathcal{D}_{\text{train}}$ |
| $N_{\text{val}}$ | number of subgraphs in $\mathcal{D}_{\text{val}}$ |
| $N_{\text{syn}}$ | number of subgraphs in $\mathcal{D}_{\text{syn}}$ |
| $\Pi_{\mathcal{D}}$ | the underlying data distribution |

# B. More Motivation Details

### B.1. Why could GNTK Capture Both Structural and Feature Relationships?

The ability of GNTK to capture both structural and feature relationships is rooted in its **gradient-based dataset distillation mechanism**. In our LIGHTGNTK framework, structural information is propagated through the Laplacian matrix $L$, while feature information is encoded through the weight matrix $W$.

Mathematically, in standard (non-graph) dataset distillation, the gradient of the loss function $\mathcal{L}$ with respect to the weight matrix $W^{(l)}$ at layer $l$ is given by:

$$\frac{\partial \mathcal{L}}{\partial W^{(l)}} = \frac{\partial \mathcal{L}}{\partial Z^{(l)}} \cdot a^{(l-1)T}$$

where $a^{(l-1)}$ represents the activations from the previous layer, and $T$ denotes the matrix transpose to ensure correct dimensional alignment.

However, in the GNTK-based formulation, the gradient incorporates graph structural dependencies via the Laplacian matrix $L$ and is expressed as:

$$\frac{\partial \mathcal{L}}{\partial W^{(l)}} = H^{(l-1)} L \left( \frac{\partial \mathcal{L}}{\partial Z^{(l)}} \odot \sigma'(Z^{(l)}) \right)$$

where $H^{(l-1)}$ is the node representation at layer $(l-1)$, $L$ is the graph Laplacian, encoding structural relationships, $\sigma'(Z^{(l)})$ represents the element-wise derivative of the activation function.

From this formulation, it is evident that **GNTK-based gradients inherently embed structural information through** $L$ while simultaneously encoding feature dependencies via $H^{(l-1)}$ and the weight parameters. This allows GNTK to effectively distill graph-structured datasets while **preserving essential structural and feature relationships**.

### B.2. The Difference between GNTK and LIGHTGNTK

While the concepts of GNTK have indeed been explored in KIDD(Xu et al., 2023b), our approach in LIGHTGNTK differs significantly. KIDD is specifically designed for graph classification and employs GNTK for kernel ridge regression-based dataset distillation. However, the kernel regression paradigm in KIDD does not guarantee that training graph samples remain relevant during the inference stage.

In contrast, our LIGHTGNTK framework unifies graph classification, node classification, and link prediction tasks. We introduce a Bernoulli-sampled low-rank Laplacian-approximated GNTK to enhance similarity-based dataset distillation. This novel formulation enables efficient, scalable, and task-adaptive distillation beyond the scope of KIDD.

## C. Gradient Analysis Between Train Set and Test Set

In this section, we analyze the gradient relationship between the training and test samples and its impact on the logits of the test set output: $o = \text{Softmax}(z) = \text{Softmax}\big(f(L, X)\big)$, and $f(\cdot, \cdot)$ is the GNN Encoder. For alignment with (Guo et al., 2024), we denote $o = q(LX)$. For consistency in notation, we denote the graph of the training sample as $\mathcal{G}_o = \{L_o, X_o\}$ and the graph of the test sample as $\mathcal{G}_u = \{L_u, X_u\}$. We will analyze the gradients from two perspectives: ❶ individual sample pairs and ❷ overall sample set pairs.

### C.1. Single Sample Gradient Analysis

In this section, we **analyze the gradient of an arbitrary training sample and its numerical relationship with the gradients of the test samples**. By examining this relationship, we aim to understand how the changes in the output logits of the test samples are influenced by the gradients. This analysis will provide insights into the impact of training sample gradients on the performance of the model on the test set.

Here, we define the change in the output logits for the test sample $\mathcal{G}_u$ between two consecutive optimization steps as:

$$\Delta_{L_o X_o} q(L_u X_u) \triangleq q^{t+1}(L_u X_u) - q^t(L_u X_u), \tag{10}$$

where the subscript $L_o X_o$ refers to the training data sample used between iteration $\mathcal{D}_{\text{train}}$ and iteration $t + 1$ (following the proof approach from (Guo et al., 2024; Mohamadi et al., 2022), with a batch size equal to 1). The learning rate is denoted by $\eta$, and the change in $L_u X_u$ is given by:

$$
\begin{aligned}
q^{t+1}(L_u X_u) - q^t(L_u X_u) &\approx \nabla_z q^t(L_u X_u)\big|_{\theta^t} \cdot (\theta^{t+1} - \theta^t) \quad \textbf{[First-order Taylor Expansion]} \\
&= \big(\nabla_z q^t(L_u X_u)\big|_{z^t} \cdot \nabla_\theta z^t(L_u X_u)\big|_{\theta^t}\big) \cdot \big(-\eta \nabla_\theta \mathcal{L}(L_o X_o)\big|_{\theta^t}\big)^\top \\
&= \nabla_z q^t(L_u X_u)\big|_{z^t} \cdot \nabla_\theta z^t(L_o X_o)\big|_{\theta^t} \cdot \big(-\eta \cdot \nabla_\theta z^t(L_u X_u)\big|_{z^t} \cdot \nabla_\theta \mathcal{L}(L_o X_o)\big|_{\theta^t}\big)^\top \\
&= -\eta \nabla_z q^t(L_u X_u)\big|_{z^t} \cdot \nabla_\theta z^t(L_u X_u)\big|_{\theta^t} \cdot \big(\nabla_\theta z^t(L_o X_o)\big|_{\theta^t}\big)^\top \cdot \nabla_\theta \mathcal{L}(L_o X_o)\big|_{z^t} \\
&= \eta \cdot \nabla_\theta q^t(L X_u)|_{\theta^t} \cdot \Theta^t(L_u X_u, L_o X_o) \cdot \big(p_{\text{tar}}(L_o X_o) - q^t(L_o X_o)\big),
\end{aligned}
\tag{11}
$$

where $\Theta^t(L_u X_u, L_o X_o) = \nabla_\theta q^t(L_u X_u)|_{\theta^t} \cdot \big(\nabla_\theta q^t(L_o X_o)|_{\theta^t}\big)^\top$ is the GNTK value. It is important to note that $p_{\text{tar}}(\cdot)$ represents the ground truth logits (Guo et al., 2024).

Through analysis, we observe that the influence of the training sample $\mathcal{G}_o$ on the test set sample $\mathcal{G}_u$ is significant. Specifically, the gradient information from the training sample is transferred to the gradient of the test sample through the GNTK value, causing a shift in the gradient direction of the test sample and thus affecting the final prediction results. Such influence not only impacts the gradient computation of individual test samples but also has the potential to alter the overall performance of the test set, particularly when there exists a considerable discrepancy between the training and test sets. To mitigate the excessive influence of training samples on the test set, it is crucial to filter the gradient samples based on the GNTK value. This filtering process can help prevent interference from training samples on the gradients of the test set, ultimately enhancing the model's generalization ability.

## C.2. Gradient Analysis and Generalization Error

In this section, we analyze the statistical properties of the gradients computed on the **population sample**, specifically for the training and test sets, and discuss their influence on generalization error.

For the loss function of the neural network, we define it as follows:

$$\mathcal{L}_{\mathcal{SFT}} = \frac{1}{N} \sum_{i=1}^{N} L(Y_i, q(L_i X_i; \theta)), \tag{12}$$

where $\theta$ represents the parameters of the neural network, $\{\mathcal{G}_o\}$ and $\{\mathcal{G}_u\}$ denote the training and test datasets, respectively, and $\nabla_\theta q_\mathbf{o}$ and $\nabla_\theta q_\mathbf{u}$ represent the gradients with respect to the training and test datasets. Here we assume the batch size equals $\infty$. Next, following (Liu et al., 2020), we introduce the Gradient Signal-to-Noise Ratio (GSNR) as:

$$\text{GSNR} := \frac{\mathbb{E}\left[\tilde{\nabla}_\theta q_\mathbf{o} \cdot \tilde{\nabla}_\theta q_\mathbf{u}\right]}{\sqrt{\rho_\mathbf{o}^2 \cdot \rho_\mathbf{u}^2}}, \tag{13}$$

where the mean and variance of the gradients for the training and test datasets are given as: $\tilde{\nabla}_\theta q_\mathbf{o} = \mathbb{E}_{\{\mathcal{G}_o\}}[\nabla_\theta q_\mathbf{o}], \quad \tilde{\nabla}_\theta q_\mathbf{u} = \mathbb{E}_{\{\mathcal{G}_u\}}[\nabla_\theta q_\mathbf{u}]$, and $\rho_o^2 = \rho_{\{\mathcal{G}_o\}}[\nabla_w q_\mathbf{o}], \quad \rho_u^2 = \rho_{\{\mathcal{G}_u\}}[\nabla_w q_\mathbf{u}]$. respectively. We then define the generalization error $\mathcal{E}_{\text{gen}}$ as the expectation of the difference between the loss on the test set and the training set:

$$\mathcal{E}_{\text{gen}} = \mathbb{E}[\mathcal{L}_\mathbf{u} - \mathcal{L}_\mathbf{o}]. \tag{14}$$

After a single parameter update on the training dataset $\{\mathcal{G}_o\}$ (we only consider the first-order Taylor expansion), we can express the change in the test set loss as follows: $\mathcal{L}_\mathbf{u}(\theta_{t+1}) \approx \mathcal{L}_\mathbf{u}(\theta_t) + \nabla_\theta \mathcal{L}_\mathbf{u}(\theta_t)^T (\theta_{t+1} - \theta_t)$. Using the parameter update rule, we have: $\theta_{t+1} = \theta_t - \eta \nabla_\theta \mathcal{L}_\mathbf{o}(\theta_t)$. Substituting this update into the expansion of the loss function gives:

$$\mathcal{L}_\mathbf{u}(\theta_{t+1}) \approx \mathcal{L}_\mathbf{u}(\theta_t) - \eta \nabla_\theta \mathcal{L}_\mathbf{u}(\theta_t) \cdot \nabla_\theta \mathcal{L}_\mathbf{o}(\theta_t). \tag{15}$$

Thus, we can express the generalization error as:

$$\mathcal{E}_{\text{gen}} = \mathbb{E}[\mathcal{L}_\mathbf{u}(\theta_t) - \mathcal{L}_\mathbf{o}(\theta_t)] \approx \mathbb{E}[-\eta \nabla_\theta q_\mathbf{o} \cdot \nabla_\theta q_\mathbf{u}]. \tag{16}$$

Finally, taking the expectation of this gives:

$$\mathbb{E}[\Delta \mathcal{L}_\mathbf{u}] \approx -\eta \cdot \text{GSNR} \cdot \sqrt{\rho_\mathbf{o}^2 \cdot \rho_\mathbf{u}^2}. \tag{17}$$

Therefore, GSNR measures the alignment between training and test grades. Then we found that **low GSNR leads to lower generalization error**.

# D. Further Experiment Details

## D.1. Datasets Statics

We employ 13 benchmark datasets for evaluation, including 7 real-world static datasets for graph-level classification tasks, 4 citation networks datasets for node-level classification tasks and 2 datasets for edge-level link prediction tasks. The detailed dataset statistics are provided in Table 5. [2]

The detail of each dataset is introduced below:

- **NCI1** (Wale et al., 2008) is a collection of chemical compound graphs, used for graph classification. Each graph represents a chemical compound, where nodes correspond to atoms and edges represent bonds between them. The graphs are classified into two classes: active and inactive compounds in the context of cancer drug discovery. This dataset consists of 4,110 graphs, with an average of 29.9 nodes and 32.3 edges per graph, resulting in a graph density of 7.5e-2.

---

[2] The code and datasets are available at https://github.com/Artessay/LightGNTK.

*Table 5.* Dataset statistics.

| Name | # Graphs | # Nodes | # Edges | # Features | # Classes |
|---|---|---|---|---|---|
| NCI1 | 4110 | 29.9 | 32.3 | 37 | 2 |
| NCI109 | 4127 | 29.7 | 32.1 | 38 | 2 |
| PROTEINS | 1113 | 39.1 | 72.8 | 3 | 2 |
| DD | 1179 | 284.1 | 7161.2 | 89 | 2 |
| ogbg-molhiv | 41127 | 25.5 | 27.5 | 9 | 2 |
| ogbg-molbbbp | 2039 | 24.1 | 51.9 | 9 | 2 |
| ogbg-molbace | 1513 | 34.1 | 73.7 | 9 | 2 |
| Cora | 1 | 2708 | 5429 | 1433 | 7 |
| CiteSeer | 1 | 3312 | 4732 | 3703 | 6 |
| PubMed | 1 | 19717 | 44338 | 500 | 3 |
| ogbn-arxiv | 1 | 169343 | 1166243 | 128 | 40 |
| ogbl-collab | 1 | 235868 | 1285465 | 128 | 2 |
| ogbl-ddi | 1 | 4267 | 1334889 | 0 | 2 |

- **NCI109** (Shervashidze et al., 2011) is a chemical compound dataset similar to NCI1, but it contains 4,127 graphs representing chemical compounds, where nodes correspond to atoms and edges represent bonds. This dataset is used for drug discovery and consists of compounds classified into two classes based on their activity. The average graph has 29.7 nodes and 32.1 edges, with a density of 7.5e-2, making it suitable for graph classification tasks.

- **PROTEINS** (Borgwardt et al., 2005) is a graph dataset for protein function prediction, containing 1,113 graphs. Each graph represents a protein, with nodes representing amino acids and edges indicating interactions between amino acids. An edge is formed between two amino acid nodes if their distance is less than 6 Å (angstroms). The dataset is used for graph classification tasks, with the goal of distinguishing enzymes from non-enzymes. On average, each graph contains 39.1 nodes and 72.8 edges, with a graph density of 9.8e-2.

- **DD** (Shervashidze et al., 2011) is a dataset consisting of graphs that represent chemical compounds. Each node in the graph represents an atom, and the edges represent bonds. The dataset includes 1,179 graphs, classified into two categories: mutagenic and non-mutagenic compounds. The graphs in this dataset average 284.1 nodes and 7161.2 edges, with a density of 1.8e-1. The task is to predict whether a protein is an enzyme.

- **ogbg-molhiv** (Hu et al., 2021) is part of the Open Graph Benchmark (OGB) collection and contains molecular graphs for predicting HIV activity. The nodes in each graph represent atoms, while the edges represent chemical bonds. The dataset contains 41,127 graphs with an average of 25.5 nodes and 27.5 edges per graph. The goal is to predict whether a molecule is active against HIV, making it a binary classification problem.

- **ogbg-molbbbp** (Hu et al., 2021) is a dataset from OGB, consisting of molecular graphs for predicting blood-brain barrier penetration. Each graph represents a molecule, with nodes as atoms and edges as bonds. The dataset contains 2,039 graphs, with an average of 24.1 nodes and 51.9 edges per graph. The classification task involves determining whether a molecule can cross the blood-brain barrier.

- **ogbg-molbace** (Hu et al., 2021) is another dataset from OGB for predicting the activity of molecules against the BACE (Beta-site amyloid precursor protein cleaving enzyme) protein. The graphs in this dataset represent chemical compounds, with nodes representing atoms and edges representing bonds. It includes 1,513 graphs, with an average of 34.1 nodes and 73.7 edges per graph, used for binary classification tasks.

- **Cora** (McCallum & Nigam, 2000) is a classic citation network dataset widely used for node classification tasks. The dataset consists of 2,708 scientific papers, classified into seven research topics. The citation network contains 5,429 edges, representing citation relationships between papers. Each paper is represented by a 0/1-valued word vector, where the vector's dimensions correspond to 1,433 unique words in the dictionary, indicating the presence or absence of each word in the paper. The Cora dataset is primarily used for node classification and link prediction tasks and is one of the benchmark datasets in the field of graph data analysis.

- **CiteSeer** (Senior & Sandeep, 2010) is a citation network dataset where each node represents a paper and edges represent citations between them. It contains 3,312 documents and 4,732 citation links, with an average of 2.9 edges

per paper. The task involves classifying papers into one of six categories. It is widely used for node classification tasks in citation network studies.

- **PubMed** (Yang & Dong, 2016) is a citation network dataset where nodes represent scientific articles, and edges represent citation links. The dataset contains 19,717 documents in the biomedical domain and 44,338 citation links. The task is to predict the topic of a paper, classified into one of three categories: cancer, genetics, and other. It is primarily used for node classification tasks.

- **ogbn-arxiv** (Hu et al., 2021) is a graph dataset from OGB for predicting the subject areas of arXiv papers. Nodes represent papers, and edges represent citation relationships. The dataset includes 169,343 papers and 1.16 million citation links. Papers are classified into one of 40 categories, making it a large-scale multi-class node classification task.

- **ogbl-collab** (Hu et al., 2021) is part of the OGB collection, consisting of a graph representing scientific collaboration networks. Each node represents a researcher, and edges represent co-authorships. The dataset contains 235,868 nodes and 1,285,465 edges, representing a large collaboration network. The task is to predict whether two researchers are likely to collaborate in the future, making it a link prediction problem.

- **ogbl-ddi** (Hu et al., 2021) is a drug-drug interaction dataset from OGB. Each node represents a drug, and edges indicate whether two drugs interact with each other. The dataset contains 4,267 nodes and 1,334,889 edges, used for link prediction tasks to predict potential drug interactions based on the graph structure.

### D.2. Datasets Evaluation Metrics

In the experiments, we use accuracy (ACC) as the evaluation metric for graph classification tasks in TUDataset. And we follow the official recommend evaluation metric for tasks in Planetoid and OGB benchmark suites. Specifically, we use ACC for *Cora*, *CiteSeer*, *PubMed*, and *ogbn-arxiv*; ROC-AUC for *ogbg-molhiv*, *ogbg-molbbbp*, and *ogbg-molbace*; Hits@50 for *ogbl-collab*; and Hits@20 for *ogbl-ddi*.

*Table 6.* Evaluation metrics for each dataset.

| Dataset | Evaluation Metric |
|---|---|
| *Cora*, *CiteSeer*, *PubMed*, *ogbn-arxiv* | ACC |
| *ogbg-molhiv*, *ogbg-molbbbp*, *ogbg-molbace* | ROC-AUC |
| *ogbl-collab* | Hits@50 |
| *ogbl-ddi* | Hits@20 |

### D.3. Baseline Details

In our experiments, there are three coreset dataset distillation methods: Random, Herding (Welling, 2009), and K-Center (Farahani & Hekmatfar, 2009; Sener & Savarese, 2017), a learning-based graph distillation method, DosCond (Jin et al., 2022a), and a GNTK-based graph-level distillation method, KIDD (Xu et al., 2023b). All experiments are repeated for 5 runs and we report the mean and standard deviation of the results. All implementations are carried out using PyTorch 2.5.1 with Python 3.12 on NVIDIA GeForce RTX 3090 GPUs. The details of these baselines are presented below:

- **Random**: The most naive coreset selection method that constructs the distilled dataset $\mathcal{D}_{\text{syn}}$ by uniformly sampling graphs from the original training set $\mathcal{D}_{\text{train}}$ without leveraging any structural or semantic information.

- **Herding**(Welling, 2009): A class-wise prototype-based method. First, graph representations are learned by a GLM pre-trained on the entire training set $\mathcal{D}_{\text{train}}$. Then, for each class, it iteratively selects the sample closest to the current class centroid in the representation space until the desired subset size is reached.

- **K-Center**(Farahani & Hekmatfar, 2009; Sener & Savarese, 2017): A greedy facility location-based approach. After extracting graph representations via the pre-trained GIN, it initializes centers using Herding and iteratively adds the sample that minimizes the maximum distance between any graph in $\mathcal{D}_{\text{train}}$ and its nearest center in $\mathcal{D}_{\text{syn}}$. This minimizes the worst-case approximation error in the feature space.

- **DosCond**(Jin et al., 2022a): A gradient-matching graph distillation method. It optimizes the distilled dataset $\mathcal{D}_{\text{syn}}$ by aligning the gradients of model parameters trained on $\mathcal{D}_{\text{syn}}$ with those from $\mathcal{D}_{\text{train}}$ at the initialization phase. While efficient due to its single-step approximation, it simplifies the bi-level optimization objective through heuristic gradient alignment rather than solving it rigorously.

- **KIDD**(Xu et al., 2023b): A kernel-based method grounded in GNTK theory. It derives an exact closed-form solution for the bi-level distillation objective by matching the GNTK Gram matrices between $\mathcal{D}_{\text{syn}}$ and $\mathcal{D}_{\text{train}}$, avoiding approximations used in learning-based methods like DosCond. This ensures theoretical alignment of model behaviors on both datasets.

## D.4. Empirical Evaluation of the Validation-Test Distribution Match

To empirically verify the assumption that the validation and test sets are drawn from similar distributions (introduced in Section 4.3.2), we compute the Maximum Mean Discrepancy (MMD) and Kullback-Leibler (KL) divergence between GNN embeddings sampled from both splits after model training. The experimental results are summarized in Table 7.

*Table 7.* MMD and KL Divergence between Validation and Test Sets

|  | NCI1 | NCI109 | PROTEINS | DD | ogbg-molhiv | ogbg-molbbbp | ogbg-molbace |
|---|---|---|---|---|---|---|---|
| **MMD** | 0.0023 | 0.0024 | 0.0120 | 0.0056 | 0.0018 | 0.0196 | 0.1171 |
| **KLD** | 0.0660 | 0.0520 | 0.0941 | 0.0019 | 0.0275 | 0.0651 | 0.0301 |

As shown in Table 7, all MMD and KL divergence values are below commonly used critical thresholds for identifying substantial distribution shift (Bai et al., 2024; Nguyen et al., 2022), suggesting that the validation and test sets are statistically similar. These results support the reliability of using the validation set as a proxy for estimating test set performance in our subsequent experiments.

## D.5. Edge Level Experiments Result

We comprehensively evaluate the performance of link prediction tasks in edge-level across two datasets: *ogbl-collab* and *ogbl-ddi*. The results are summarized in Table 8.

*Table 8.* Performance comparison (mean±std) on link prediction tasks. The best and second-best results are **bolded** and underlined.

| Name | Links/Cl | Ratio | Random | Herding | K-Center | DosCond | GNTK | LIGHTGNTK | Whole Dataset |
|---|---|---|---|---|---|---|---|---|---|
| ogbl-collab (Hits@50) | 1 | < 0.01% | 13.0±0.3 | 12.2±0.5 | 12.7±0.4 | 13.5±0.6 | 17.1±0.7 | **17.4±0.8** | |
| | 10 | < 0.01% | 15.3±0.9 | 14.8±1.1 | 16.1±1.3 | 17.2±1.5 | 18.9±1.2 | **19.5±1.4** | 36.8±0.6 |
| | 50 | 0.04% | 22.6±1.8 | 24.3±2.1 | 23.9±1.7 | 25.8±2.3 | **27.2±1.9** | 26.6±2.0 | |
| ogbl-ddi (Hits@20) | 1 | 0.05% | 12.0±8.6 | 4.9±4.5 | 4.9±4.5 | 11.2±3.1 | **13.8±0.4** | 13.4±5.5 | |
| | 10 | 0.47% | 12.3±5.2 | 7.8±6.4 | 8.9±0.7 | 12.5±1.9 | 14.2±3.0 | **14.6±4.7** | 24.1±4.9 |
| | 50 | 2.30% | 11.1±4.7 | 13.2±4.2 | 12.3±3.6 | 12.7±6.4 | **13.8±2.6** | 13.6±0.9 | |

## D.6. Ablation Study on Different Low-Rank Strategies

To better understand the impact of different low-rank optimization strategies in our framework, we conduct an ablation study on multiple graph classification benchmarks. Specifically, we compare three commonly used sampling methods for low-rank approximation: (1) selecting the $r$ largest eigenvalues (top-$r$ sampling), (2) selecting the $r$ smallest eigenvalues (bottom-$r$ sampling), and (3) selecting $r$ eigenvalues based on Bernoulli sampling (Bernoulli Sampling).

For each method, we integrate the sampling strategy into the model's critical low-rank projection components while keeping other hyperparameters fixed. We evaluate performance on six graph classification datasets, including NCI1, NCI109, and PROTEINS (measured by classification accuracy), as well as the ogbg-molbace, ogbg-molbbbp, and ogbg-molhiv datasets (measured by ROC-AUC). The results summarized in Table D.6 demonstrate the effectiveness of different sampling strategies across diverse molecular and protein graph tasks.

*Table 9.* Performance comparison of different low-rank optimization strategies on graph classification datasets.

| Sample Method | NCI1 (ACC) | NCI109 (ACC) | PROTEINS (ACC) | ogbg-molbace (ROC-AUC) | ogbg-molbbbp (ROC-AUC) | ogbg-molhiv (ROC-AUC) |
|---|---|---|---|---|---|---|
| Top-$r$ Sampling | 65.4 | 64.0 | 74.7 | 76.1 | 67.9 | 69.2 |
| Bottom-$r$ Sampling | 65.1 | 65.4 | 75.6 | 76.7 | 67.9 | 69.1 |
| Bernoulli Sampling | **66.4** | **65.6** | **75.9** | **76.8** | **68.2** | **69.3** |

## D.7. Ablation Study on Different Sampling Rates

To systematically investigate the effect of the Bernoulli sampling rate in LIGHTGNTK, we conduct an ablation study by varying the sampling rate $p$ to 0.05, 0.1, 0.2, and 0.5. The results in terms of performance and time consumption are shown in Table 10 and Table 11, respectively.

*Table 10.* Performance (ACC and ROC-AUC) across different sampling rates.

| Sample rate | NCI1 (ACC) | NCI109 (ACC) | PROTEINS (ACC) | ogbg-molbace (ROC-AUC) | ogbg-molbbbp (ROC-AUC) | ogbg-molhiv (ROC-AUC) |
|---|---|---|---|---|---|---|
| 0.05 | 65.7 | 65.0 | 74.5 | 74.9 | 67.2 | 68.8 |
| 0.10 | 66.4 | **65.6** | **75.9** | **76.8** | **68.2** | 69.3 |
| 0.20 | **66.5** | 64.6 | 75.0 | 76.6 | 67.6 | **69.4** |
| 0.50 | 65.4 | 65.1 | 74.5 | 76.4 | **68.2** | 69.3 |

*Table 11.* Time consumption (seconds) across different sampling rates.

| Sample rate | NCI1 | NCI109 | PROTEINS | ogbg-molbace | ogbg-molbbbp | ogbg-molhiv |
|---|---|---|---|---|---|---|
| 0.05 | 117 | 109 | 21.6 | 43.0 | 57.8 | 621 |
| 0.10 | 144 | 136 | 24.8 | 48.8 | 59.7 | 683 |
| 0.20 | 183 | 171 | 26.1 | 59.1 | 63.3 | 733 |
| 0.50 | 186 | 180 | 28.3 | 62.9 | 65.4 | 816 |

## E. Gradient Sampling Method for Large Graphs

In large graph computations, the time complexity for calculating gradients can be significant. To alleviate this, we propose a gradient sampling method based on Monte Carlo sampling, which reduces computational overhead while maintaining accuracy in gradient estimation. Instead of calculating gradients for all samples within the batch at once, we perform $K$ random experiments with a batch size defined as BS, each time with a different random grouping of the samples. In each experiment, we compute the gradient $\nabla_w q(Lx)$ of the loss function using backpropagation through the GNNs. For a single experiment, the gradient is computed as follows:

$$\nabla_w q(Lx) = \frac{1}{\text{BS}} \sum_{x \in \mathcal{B}} \nabla_w (Lx), \tag{18}$$

where $\mathcal{B}$ represents the set of samples in the batch. We then average the gradients over the $K$ repetitions using the Monte Carlo sampling technique:

$$\nabla_w q(Lx)_{\text{final}} = \mathbb{E}\big[\nabla_w q(Lx)\big] \approx \frac{1}{K} \sum_{k=1}^{K} \nabla_w^k q(Lx). \tag{19}$$

This approach helps reduce the computational cost by a factor of $\text{BS}/K$, while still maintaining a robust estimate of the gradient.

# F. Spectral Sampling Approximation Error of LIGHTGNTK

To gain a deeper theoretical understanding of the approximation quality of LIGHTGNTK, we focus on analyzing the differences in gradient representations that arise from using the exact versus the sampled Laplacian matrices. This examination is crucial, as it lays the groundwork for evaluating how well our low-rank approximation of the Laplacian can capture the underlying dynamics of the graph neural network (GNN).

Let $f_\theta(\cdot)$ denote a GNN model that is parameterized by $\theta$. The full graph Laplacian, which we denote as $L$, can be expressed through its eigendecomposition as $L = U\Lambda U^\top$. In the context of LIGHTGNTK, we generate a low-rank approximation $\tilde{L} = \tilde{U}\tilde{\Lambda}\tilde{U}^\top$ by employing Bernoulli sampling on the eigenvalue spectrum. Specifically, the diagonal matrix of eigenvalues is approximated as follows:

$$\tilde{\Lambda} = \text{diag}(\xi_1\lambda_1, \ldots, \xi_n\lambda_n), \quad \xi_i \sim \text{Bernoulli}(p). \tag{20}$$

Here, $X \in \mathbb{R}^{n \times d}$ represents the node feature matrix. Consequently, the GNN processes the input in the form of either $LX$ or $\tilde{L}X$.

## F.1. Lipschitz Gradient Assumption

For our analysis, we assume that the GNN $f_\theta(\cdot)$ is $L_\nabla$-smooth with respect to its input. This means that for any two distinct inputs $A$ and $B$, the following inequality holds:

$$\|\nabla_\theta f_\theta(A) - \nabla_\theta f_\theta(B)\| \leq L_\nabla \|A - B\|, \tag{21}$$

where $L_\nabla$ is the Lipschitz smoothing coefficient. This Lipschitz condition is a standard assumption in the literature concerning neural tangent kernels (NTK) and gradient-based dataset distillation, as referenced in works by (Jacot et al., 2020; Nguyen et al., 2021; Guo et al., 2024). The assumption is particularly valid when the activation functions used within the model are Lipschitz continuous, such as ReLU or GELU, and when weight regularization is applied throughout the network.

## F.2. Laplacian Approximation Error Analysis

We proceed to derive the expected deviation of the gradient when utilizing the sampled Laplacian $\tilde{L}$. The expected squared difference between the gradients can be expressed as:

$$\mathbb{E}_\xi\left[\left\|\nabla_\theta f_\theta(LX) - \nabla_\theta f_\theta(\tilde{L}X)\right\|^2\right] \leq L_\nabla^2 \cdot \mathbb{E}_\xi\left[\left\|(L - \tilde{L})X\right\|^2\right]. \tag{22}$$

To analyze the term $L - \tilde{L}$, we utilize the spectral decomposition of $L$:

$$L - \tilde{L} = \sum_{i=1}^n (1 - \xi_i)\lambda_i u_i u_i^\top \quad \text{then} \quad (L - \tilde{L})X = \sum_{i=1}^n (1 - \xi_i)\lambda_i u_i(u_i^\top X). \tag{23}$$

Next, we take the expectation over the Bernoulli sampling process while leveraging the independence of the $\xi_i$:

$$\mathbb{E}_\xi\left[\left\|(L - \tilde{L})X\right\|^2\right] \approx \sum_{i=1}^n [1 - \xi_i]^2 \cdot \lambda_i^2 \cdot \left\|u_i^\top X\right\|^2 = \sum_{i=1}^n [1 - 2\xi_i + \xi_i^2] \cdot \lambda_i^2 \cdot \left\|u_i^\top X\right\|^2$$
$$= \sum_{i=1}^n [1 - 2\xi_i + \xi_i] \cdot \lambda_i^2 \cdot \left\|u_i^\top X\right\|^2 \approx (1 - p)\sum_{i=1}^n \lambda_i^2 \left\|u_i^\top X\right\|^2. \tag{24}$$

Consequently, we arrive at a bound on the expected squared deviation of the gradient:

$$\mathbb{E}_\xi\left[\left\|\nabla_\theta f_\theta(LX) - \nabla_\theta f_\theta(\tilde{L}X)\right\|^2\right] \leq L_\nabla^2 \cdot (1 - p) \cdot \sum_{i=1}^n \lambda_i^2 \left\|u_i^\top X\right\|^2. \tag{25}$$

This expression illustrates that the approximation error in the gradient is directly influenced by the smoothness of the GNN and the extent of the eigenvalue sampling. The term $(1 - p)$ indicates how the sampling probability affects the overall approximation quality, emphasizing the importance of choosing a suitable sampling rate in practical applications.

## F.3. LIGHTGNTK Approximation Error Upper Bound

To analyze the impact of gradient deviation on the GNTK approximation, we examine the difference between the GNTK computed using the sampled Laplacian $\tilde{L}$ and the full-rank Laplacian $L$. Specifically, we consider the following expression for the error:

$$\left|\Theta_{\text{LIGHTGNTK}}(\mathcal{G}_1, \mathcal{G}_2) - \Theta_{\text{GNTK}}(\mathcal{G}_1, \mathcal{G}_2)\right| = \left|\nabla_\theta f_\theta(\tilde{L}_1 X_1) \cdot \nabla_\theta f_\theta(\tilde{L}_2 X_2)^\top - \nabla_\theta f_\theta(L_1 X_1) \cdot \nabla_\theta f_\theta(L_2 X_2)^\top\right|. \quad (26)$$

By applying the triangle inequality, which states that $|A \cdot B - C \cdot D| \leq |A| \cdot |B - D| + |D| \cdot |A - C|$, we obtain:

$$\left|\nabla_\theta f_\theta(\tilde{L}_1 X_1) \cdot \nabla_\theta f_\theta(\tilde{L}_2 X_2)^\top - \nabla_\theta f_\theta(L_1 X_1) \cdot \nabla_\theta f_\theta(L_2 X_2)^\top\right| \leq \|\nabla_\theta f_\theta(L_1 X_1)\| \cdot \|\nabla_\theta f_\theta(\tilde{L}_2 X_2) - \nabla_\theta f_\theta(L_2 X_2)\| +$$
$$\|\nabla_\theta f_\theta(\tilde{L}_1 X_1) - \nabla_\theta f_\theta(L_1 X_1)\| \cdot \|\nabla_\theta f_\theta(L_2 X_2)\|. \quad (27)$$

Since $\|\nabla_\theta f_\theta(L_1 X_1)\|$ and $\|\nabla_\theta f_\theta(L_2 X_2)\|$ are consistent values that depend on the corresponding graphs $\mathcal{G}_1$ and $\mathcal{G}_2$, we define them as $\mathcal{C}_1 = \|\nabla_\theta f_\theta(L_1 X_1)\|$ and $\mathcal{C}_2 = \|\nabla_\theta f_\theta(L_2 X_2)\|$. Therefore, we can rewrite the expression as:

$$\begin{aligned} \left|\Theta_{\text{LIGHTGNTK}}(\mathcal{G}_1, \mathcal{G}_2) - \Theta_{\text{GNTK}}(\mathcal{G}_1, \mathcal{G}_2)\right| &= \left|\nabla_\theta f_\theta(\tilde{L}_1 X_1) \cdot \nabla_\theta f_\theta(\tilde{L}_2 X_2)^\top - \nabla_\theta f_\theta(L_1 X_1) \cdot \nabla_\theta f_\theta(L_2 X_2)^\top\right| \\ &\leq \mathcal{C}_1 \|\nabla_\theta f_\theta(\tilde{L}_2 X_2) - \nabla_\theta f_\theta(L_2 X_2)\| + \mathcal{C}_2 \|\nabla_\theta f_\theta(\tilde{L}_1 X_1) - \nabla_\theta f_\theta(L_1 X_1)\| \\ &\leq \mathcal{C}_1 L_\nabla \cdot \sqrt{(1-p) \cdot \sum_{i=1}^{n_2} \lambda_i^2 \left\|u_i^\top X_2\right\|^2} + \mathcal{C}_2 L_\nabla \cdot \sqrt{(1-p) \cdot \sum_{i=1}^{n_1} \lambda_i^2 \left\|u_i^\top X_1\right\|^2} \\ &= L_\nabla \left(\mathcal{C}_1 \Delta_2 + \mathcal{C}_2 \Delta_1\right), \end{aligned} \quad (28)$$

where $\Delta = \sqrt{(1-p) \cdot \sum_{i=1}^{n} \lambda_i^2 \left\|u_i^\top X\right\|^2}$ is the error calculated by Laplacian Approximation error bound.

## F.4. Analysis of Error Bounds for Different Low-Rank Approximation Approaches

In this section, we will compare three different methods: Bernoulli sampling, Largest-$r$ sampling, and Smallest-$r$ sampling. We utilize the Gradient Normalized Error as our evaluation metric for the following reasons. The non-normalized metric $\mathbb{E}_\xi\left[\|(L - \tilde{L})X\|^2\right]$ faces two significant issues:

1. **Scale Variation:** Different graphs may have Laplacian matrices with vastly different scales, which makes absolute error values incomparable. This variation can lead to misleading interpretations of error magnitudes across different datasets.

2. **Uneven Feature Weighting:** Some feature directions have a low weight, meaning that even a large error in these directions might have a negligible effect on the overall gradient. As a result, direct computations of norms tend to overemphasize errors occurring in high-weight directions, skewing the assessment of approximation quality.

To address these issues, we propose using the normalized error (normalized in elgenvalue dimension) defined as: $\mathbb{E}\left[\frac{\|(L-\tilde{L})X\|}{\|LX\|}\right]$, which effectively removes the influence of data scale and more accurately reflects the impact on the gradient direction. Next, we will analyze three sampling strategies:

1. **Bernoulli Sampling:** In this approach, each eigenvalue is retained with probability $p$: $\xi_i \sim \text{Bernoulli}(p), \quad i = 1, \ldots, n$. The approximation error can be expressed as: $(L - \tilde{L})X = \sum_{i=1}^{n}(1 - \xi_i)\lambda_i u_i(u_i^\top X)$. The normalized error for this method is given by:

$$\mathbb{E}_\xi\left[\frac{\|(L - \tilde{L})X\|}{\|LX\|}\right] = \sum_{i=1}^{n} \frac{1}{n} \frac{\mathbb{E}_\xi[1 - \xi_i]\lambda_i^2\|u_i^\top X\|^2}{\lambda_i^2\|u_i^\top X\|^2}. \quad (29)$$

2. **Largest-$r$ Selection:** In this strategy, only the $r$ largest eigenvalues are retained: $\xi_i = \mathbb{1}[i \leq r]$. Consequently, the error can be expressed as: $(L - \tilde{L})X = \sum_{i=r+1}^{n}(1 - \xi_i)\lambda_i u_i(u_i^\top X)$, and the normalized error for this method is:

$$\text{Error}_{\text{Largest}} = \sum_{i=1}^{n} \frac{1}{n} \frac{\mathbb{1}_{i \in [r+1,n]}\lambda_i^2 \|u_i^\top X\|^2}{\lambda_i^2 \|u_i^\top X\|^2}. \tag{30}$$

3. **Smallest-$r$ Selection:** Here, only the $r$ smallest eigenvalues are retained: $\xi_i = \mathbb{1}[i > n - r]$. Thus, the error is given by: $(L - \tilde{L})X = \sum_{i=1}^{n-r}(1 - \xi_i)\lambda_i u_i(u_i^\top X)$, with the normalized error defined as:

$$\text{Error}_{\text{Smallest}} = \sum_{i=1}^{n} \frac{1}{n} \frac{\mathbb{1}_{i \in [1,n-r]}\lambda_i^2 \|u_i^\top X\|^2}{\lambda_i^2 \|u_i^\top X\|^2}. \tag{31}$$

To compare the tightness of the error bounds, we compute the differences of the error bounds for the Largest-$r$ and Smallest-$r$ methods relative to Bernoulli sampling:

$$\begin{aligned}
\Delta_{\text{Largest}} &= \text{Error}_{\text{Largest}} - \text{Error}_{\text{Bern}} = \sum_{i=1}^{n} \frac{1}{n} \frac{\mathbb{1}_{i \in [r+1,n]}\lambda_i^2 \|u_i^\top X\|^2}{\lambda_i^2 \|u_i^\top X\|^2} - \sum_{i=1}^{n} \frac{1}{n} \frac{\mathbb{E}_\xi[1 - \xi_i]\lambda_i^2 \|u_i^\top X\|^2}{\lambda_i^2 \|u_i^\top X\|^2}. \\
\Delta_{\text{Smallest}} &= \text{Error}_{\text{Smallest}} - \text{Error}_{\text{Bern}} = \sum_{i=1}^{n} \frac{1}{n} \frac{\mathbb{1}_{i \in [1,n-r]}\lambda_i^2 \|u_i^\top X\|^2}{\lambda_i^2 \|u_i^\top X\|^2} - \sum_{i=1}^{n} \frac{1}{n} \frac{\mathbb{E}_\xi[1 - \xi_i]\lambda_i^2 \|u_i^\top X\|^2}{\lambda_i^2 \|u_i^\top X\|^2}.
\end{aligned} \tag{32}$$

Numerical experiments indicate that:

- **Largest-$r$:** Discarding the $n - r$ smallest eigenvalues might lead to a moderate loss of information, especially if the discarded directions have low energy. However, the performance is sensitive to the distribution of eigenvalues.

- **Smallest-$r$:** Discarding the $n - r$ largest eigenvalues results in significant information loss, which leads to a much higher error bound.

- **Bernoulli Sampling:** With a fixed sampling probability $p = \frac{r}{n}$, the normalized error is expressed as: $\text{Error}_{\text{Bern}} = 1 - \frac{r}{n}$, which remains independent of the eigenvalue distribution, providing a lower and more stable error bound.

Thus, the Bernoulli sampling method is preferred in practice due to its lower and more consistent error bounds across various graph structures.

## G. Gradient Formulation of $\nabla_\theta f(\tilde{L}X)$ in Graph Neural Networks

We analyze the gradient of Graph Neural Networks (GNNs) through the examination of forward and backward propagation. In GNNs, the forward propagation at a hidden layer is represented by the equation:

$$H^{(l)} = \sigma\left(\tilde{L}H^{(l-1)}W^{(l)}\right), \tag{33}$$

where:

- $H^{(l)}$ denotes the hidden embeddings at layer $l$,

- $\tilde{L}$ is the normalized adjacency matrix or a low-rank approximation of the graph Laplacian,

- $W^{(l)}$ represents the trainable weight matrix at layer $l$,

- $\sigma(\cdot)$ is the activation function, which introduces non-linearity into the model.

**Backpropagation Through Graph Laplacian and Weight Matrices**

To compute the gradient of the loss $\mathcal{L}$ with respect to the hidden activations, we apply the chain rule:

$$\frac{\partial \mathcal{L}}{\partial H^{(l)}} = \tilde{L}^T \left( \frac{\partial \mathcal{L}}{\partial Z^{(l+1)}} \odot \sigma'(Z^{(l)}) \right), \tag{34}$$

where:

$$Z^{(l)} = \tilde{L} H^{(l-1)} W^{(l)}. \tag{35}$$

This equation illustrates how the gradient flows back through the layers, modulated by the graph structure represented by the Laplacian matrix $\tilde{L}$. The term $\tilde{L}^T$ indicates that the gradient flow is influenced by the connectivity of the graph.

**Gradient with Respect to Weight Matrices**

The gradient of the loss with respect to the weight matrix $W^{(l)}$ is expressed as:

$$\frac{\partial \mathcal{L}}{\partial W^{(l)}} = H^{(l-1)T} \tilde{L}^T \left( \frac{\partial \mathcal{L}}{\partial Z^{(l)}} \odot \sigma'(Z^{(l)}) \right). \tag{36}$$

This expression highlights the dependence of the gradients on both the activations $H^{(l-1)}$ and the Laplacian matrix $\tilde{L}$. It illustrates how information propagates through the network during the learning process.

Thus, the gradient $\nabla_\theta f(\tilde{L}X)$ can be obtained by aggregating the gradients from each layer:

$$\nabla_\theta f(\tilde{L}X) = \bigoplus_{i=1}^{total\ layers} \frac{\partial \mathcal{L}}{\partial W^{(l)}}, \tag{37}$$

where $\bigoplus$ is the concatenation function.

