# OpenReview forum: "Efficient Graph Continual Learning via Lightweight Graph Neural Tangent Kernels-based Dataset Distillation"
_ICML.cc/2025/Conference — ICML 2025 poster_

### Official Review · Reviewer_YvS6 · 2025-03-05

**Overall Recommendation:** 4

**Summary:**

This paper proposed a graph dataset distillation method via Graph Neural Tangent Kenrels (GNTK) for efficient graph continual learning.

The main idea is using the Bernoulli sampling method to approximate the graph Laplacian which is required for computing the gradients.
By carefully setting the probability of Bernoulli distribution, this paper claimed that it could trade off the computational efficiency and approximation error. Further, this paper proposed a data selection and a supervised fine-tuning method to achieve the authors' goal.

The experimental results seem to support its claims.

Overall, some parts of this paper are unclear and thus I am confused by these, at least in this version.

**Claims And Evidence:**

The key claims in this paper involve two aspects: 1) Low-rank optimization; and 2) The data-selection and supervised fine-tuning(SFT).

1.1. The motivation of Bernoulli random variables.

> Suppose we have an undirected graph and its adjacency matrix $A$, and we have the corresponding graph Laplacian $L:=D-A\in\mathbb{R}^{N\times N}$ where $D$ is the degree matrix [1]. The first question is why do we need to use Bernoulli distribution? According to the Authors' claims, we just need $r$ ranks, so why not directly choose the largest $r$ eigenvalues?

> From efficiency, these two do not have any difference; From approximation error,  the largest-$r$ eigenvalues lead to a smaller approximation error than any choice of Beronulli sampling.

1.2. I am not sure why and how low-rank optimization helps efficiency.

> According to (2) and (3) in the manuscript, I find that the low-rank graph Laplacian $\tilde{L}$ is still $N\times N$.

> To compute the $\tilde{L}$, one need to have the original graph Laplacian $L$; And then apply the spectral decompostion on it $L=U \Lambda U^\top$; Next, computing its low-rank approximation $\tilde{L}$; Finally, Replace $L$ with $\tilde{L}$. I am confused by it. Why not directly use $L$ since $L$ and $\tilde{L}$ have the same shape?

2. Data selection and SFT.

> The definition of $\Theta(G_i)$ in (4) is unclear. In (3) $\Theta(G_1, G_2)$ is a kernel function with two inputs; In Page 6, Line 326, $\Theta$ is a $N\times M$ matrix; But in (4), $\Theta(G_i)$ becomes a values with a single input.

> The motivation of (5) is unclear. I admit that my knowledge in the field of SFT may be insufficient. But I failed to understand what (5) was doing. I suggest the authors to explain it more.


[1] https://en.wikipedia.org/wiki/Laplacian_matrix

**Essential References Not Discussed:**

None.

**Experimental Designs Or Analyses:**

None.

**Methods And Evaluation Criteria:**

Same as ``Claims and Evidence''.

**Other Comments Or Suggestions:**

1. Page 5, line 239. Formally writing a theorem or proposition before the proof may be better than directly writing the proof.

2. Late error. Page 18, line 946.

**Other Strengths And Weaknesses:**

None.

**Questions For Authors:**

None.

**Relation To Broader Scientific Literature:**

None.

**Theoretical Claims:**

Seem to be correct.

---

> ### Author Rebuttal · Authors · 2025-03-30
>
> ## Q1: Why use Bernoulli distribution than largest r eigenvalues?
>
> We adopt Bernoulli sampling because it **effectively balances high-frequency and low-frequency components, preserving both local and global graph structures**. In contrast, **directly selecting the largest $r$ eigenvalues mainly captures local variations, while selecting the smallest $r$ eigenvalues emphasizes global structures.**
>
> ### **1. Theoretical Justification**
>
> Under Lipschitz gradient assumption [1], we assume that the GNN $ f_\theta $ satisfies
> $$
> \|\nabla_\theta f_\theta(A) - \nabla_\theta f_\theta(B)\| \leq L_\nabla \|A - B\|,
> $$
> $L_\nabla$ is the constant. Therefore, when approximating $ L $ with $ \tilde{L} $, the induced error in the gradients $ \| \nabla f_\theta(LX) - \nabla f_\theta(\tilde{L} X) \| $ is controlled by $ \| f_\theta(LX) - f_\theta(\tilde{L} X) \| $. Given eigen-decomposition:
> $
> L = U \Lambda U^\top, \tilde{L} = \tilde{U} \tilde{\Lambda} \tilde{U}^\top,
> $
> where $ \tilde{L} $ is a low-rank approximation, we introduce a selection indicator $ \xi_i $ for each eigenvalue $ \lambda_i $, leading to an error formulation:
> $$
> (L-\tilde{L})X = \sum_{i=1}^{n}(1-\xi_i)\lambda_i u_i (u_i^\top X).
> $$
>
> To ensure a fair comparison, we analyze the **normalized error bound**:
> $$
> \mathbb{E}_\xi\left[\frac{\|(L-\tilde{L})X\|}{\|LX\|}\right] =  \frac{\sum \mathbb{E}[1-\xi_i]\ \lambda_i^2\|u_i^\top X\|^2}{\sum \lambda_i^2\|u_i^\top X\|^2}
> $$
>
> For Bernoulli, where $ \xi_i \sim \text{Bernoulli}(p) $ with $ p = \frac{r}{n} $, $
> \mathbb{E}[1-\xi_i] = 1-p \approx 1-\frac{r}{n}. $
>
> Thus, the normalized error bound is:
> $
> \text{Error}_{\text{Bern}} \approx 1 - \frac{r}{n}.
> $ This result is independent of the eigenvalue distribution, ensuring robustness across different graphs.
>
> In comparison,
> $$
> \frac{\sum_{i=r+1}^{n}\lambda_i^2\|u_i^\top X\|^2}{\sum_{i=1}^{n}\lambda_i^2\|u_i^\top X\|^2} = \text{Error}_{\text{Largest}}.
> $$
>
> $$
> \frac{\sum_{i=1}^{n-r}\lambda_i^2\|u_i^\top X\|^2}{\sum_{i=1}^{n}\lambda_i^2\|u_i^\top X\|^2} = \text{Error}_{\text{Smallest}} .
> $$
>
> **These methods depend on the eigenvalue spectrum; simply dropping local or global signals will lead to significantly larger errors** [2, 3].
>
> ### **2. Experimental Validation**
>
> We conducted a comparative analysis of various sampling strategies:
>
> |Method|NCI1|NCI109|PROTEINS|molbace|molbbbp|molhiv|
> |-|-|-|-|-|-|-|
> |Bernoulli|**66.4**|**65.6**|**75.9**|**76.8**|**68.2**|**69.3**|
> |Largest|65.4|64.0|74.7|76.1|67.9|69.2|
> |Smallest|65.1|65.4|75.6|76.7|67.9|69.1|
>
> Experimental results demonstrate that **Bernoulli sampling achieves the best performance** in preserving both global and local graph information.
>
> [1]LECTURES ON LIPSCHITZ ANALYSIS.
>
> [2]Self-supervised graph-level representation learning with local and global structure. PMLR, 2021.
>
> [3]From local to global: Spectral-inspired graph neural networks. NeurIPS 2022.
>
> ---
>
> ## Q2: Explanation of low-rank optimization
>
> We acknowledge that our initial complexity analysis contained an error. Below, we provide a **corrected and more detailed explanation**.
>
> The efficiency improvement of optimization primarily stems from reducing the computational complexity of matrix operations. Instead of directly computing $N \times N$ dense matrix $ L $, we use a sampled low-rank approximation:
> $
> L \approx U_r \Lambda_r U_r^T,
> $
> where:  $ U_r \in \mathbb{R}^{N \times r},   \Lambda_r \in \mathbb{R}^{r \times r}. $
>
> To compute $ \tilde{L} X = U_r \Lambda_r U_r^T X $ efficiently, we decompose into 3 sequential steps:
> - 1.
> $
> Z = U_r^T X, \quad Z \in \mathbb{R}^{r \times d}, \quad \text{cost: } O(rNd)
> $
> - 2.
> $
> Y = \Lambda_r Z = \Lambda_r (U_r^T X), \quad Y \in \mathbb{R}^{r \times d}, \quad \text{cost: } O(r^2 d)
> $
> - 3.
> $
> \tilde{L} X = U_r Y = U_r (\Lambda_r (U_r^T X)), \quad \text{cost: } O(Nrd)
> $
>
> Thus, overall computational complexity:
> $
> O(rNd) + O(r^2 d) + O(Nrd) \approx O(Nrd),
> $
> where $ r \ll N $. This significantly reduces the complexity compared to the naive approach **$ O(N^2 d) $**. This low-rank approximation approach is conceptually related to **linear attention** in Transformers [4].
>
> [4]Transformers are RNNs: Fast Autoregressive Transformers with Linear Attention.  ICML 2020.
>
> ---
>
> ## Q3: Unclear definition of Θ(Gi)
>
> We will correct this **inconsistency in the notation of Θ in the final version of the paper** by carefully distinguishing between the kernel function and the matrix and updating notation table to ensure clarity.
>
> ---
>
> ## Q4: Definition of equation (5)
> Equation (5) is a combination of two terms:
> - Cross-entropy loss: $\frac{1}{M} \sum_{i=1}^{M} - o_i \log(\hat{o}_i)$ measures the difference between true labels $o_i$ and predicted probabilities $\hat{o}_i$.
> - Regularization: $\eta \|\Theta\|^2$, is a regularization term that adds a penalty proportional to the squared $L_2$ norm of model parameters $\Theta$ to prevent overfitting.
> ---
> ## S1&S2: Typo Errors
> We will carefully proofread and improve the manuscript. Thanks.

---

### Official Review · Reviewer_32y9 · 2025-03-09

**Overall Recommendation:** 4

**Summary:**

This work introduces a novel LIGHTGNTK that contains a low-rank GNTK approximation via Bernoulli sampling and a unified subgraph anchoring strategy for efficient and effective dataset distillation in multi-level tasks.

**Claims And Evidence:**

The main claim of the paper is that the low-rank approximation of the Laplacian matrix in GNTK can efficiently capture both structure and feature relationships in gradient-based dataset distillation. This claim is derived from the structure-based Laplacian matrix and the features-based similarity matrix.

**Essential References Not Discussed:**

N/A

**Experimental Designs Or Analyses:**

To verify the versatility of the proposed dataset distillation framework on different graph tasks, 7 graph classification datasets, 4 node classification datasets, and 2 link prediction datasets with limited training data are used in this paper to conduct comprehensive experiments.

**Methods And Evaluation Criteria:**

The proposed LIGHTGNTK and three evaluation tasks make sense for the dataset distillation at hand.

**Other Comments Or Suggestions:**

W1: Ablation studies on key components (e.g., sampling probability and layer-specific gradients) are missing.

W2: It would be better to provide the computational efficiency of LIGHTGNTK.

W3: Why some Graph Representation Learning models are not discussed in this paper? Such as:

Tan, S., Li, D., Jiang, R., Zhang, Y. and Okumura, M., 2024, July. Community-invariant graph contrastive learning. In Proceedings of the 41st International Conference on Machine Learning (pp. 47579-47606).

You, Y., Chen, T., Sui, Y., Chen, T., Wang, Z. and Shen, Y., 2020. Graph contrastive learning with augmentations. Advances in neural information processing systems, 33, pp.5812-5823.

**Other Strengths And Weaknesses:**

S1: A novel integration of low-rank GNTK approximation with dataset distillation has been proposed in the paper, enabling efficient graph continual learning.

S2: Comprehensive empirical validation across 13 datasets and multi-level tasks (node/edge/graph) have been conducted in the paper, demonstrating broad applicability in the future.

S3: The theoretical guarantees on approximation quality have been proven, enhancing methodological credibility.

**Questions For Authors:**

Q1. The ablation experiments are suggested to be conducted to evaluate the effectiveness of each proposed component in LIGHTGNTK.

Q2. It would be better to report the original training time on the whole dataset, which can validate the efficiency of this approach.

**Relation To Broader Scientific Literature:**

Downstream tasks such as graph continual learning, graph classification, node classification, link prediction, and graph foundation models will benefit.

**Theoretical Claims:**

The paper validates its theoretical claim of high-quality low-rank approximation by using Bernoulli-based sampling with a theoretical guarantee.

---

> ### Author Rebuttal · Authors · 2025-03-31
>
> We sincerely appreciate your comments, suggestions, and every effort spent on reviewing our work. Here we attempt to address all your remaining concerns. In the following, we quote your comments and then give our detailed response point-by-point.
>
> ---
>
> ## W1: Ablation studies on key components are missing.
>
> We acknowledge that our initial submission lacked comprehensive ablation studies. To rigorously validate this, we have **performed extensive ablation experiments focusing on the sampling probability and the use of layer-specific gradients**.
>
> ### 1. Sampling Probability
>
> We investigated the impact of dataset distillation quality and computation cost **under different sampling rates** for Bernoulli sampling:
>
> Table 1.  Performance Comparison among Different Sampling Rates
> | sample rate | NCI1(ACC) | NCI109(ACC) | PROTEINS(ACC) | ogbg-molbace(ROC-AUC) | ogbg-molbbbp(ROC-AUC) | ogbg-molhiv(ROC-AUC) |
> |---|---|---|---|---|---|---|
> | 0.05 | 65.7 | 65.0 | 74.5 | 74.9 | 67.2 | 68.8 |
> | 0.1 | 66.4  | **65.6** | **75.9** | **76.8** | **68.2** | 69.3 |
> | 0.2 | **66.5** | 64.6 | 75.0 | 76.6 | 67.6 | **69.4** |
> | 0.5 | 65.4 | 65.1 | 74.5 | 76.4 | **68.2**  | 69.3 |
>
> Table 2.  Time consumption (seconds) among Different Sampling Rates
> | sample rate  | NCI1 | NCI109 | PROTEINS | ogbg-molbace | ogbg-molbbbp | ogbg-molhiv |
> |---|---|---|---|---|---|---|
> | 0.05| 117 | 109 | 21.6 | 43.0 | 57.8 | 621 |
> | 0.1  | 144 | 136 | 24.8 | 48.8 | 59.7 | 683 |
> | 0.2  | 183 | 171 | 26.1 | 59.1 | 63.3 | 733 |
> | 0.5  | 186 | 180 | 28.3 | 62.9 | 65.4 | 816 |
>
> **The results indicate that a sampling probability of 0.1 achieves the best balance between training efficiency and model accuracy.**
> Lower probabilities leed to performance decay due to inadequate training data, while higher probabilities increase computational cost without significant gains in performance.
>
> ### **2. Layer-Specific Gradients**
>
> We analyzed the effect of employing layer-specific gradients by comparing it with LIGHTGNTK that uses gradients across all layers, first layer and last layer.
>
> Table 3.  Performance of gradient computation using layer-specific gradients
> | dataset  | NCI1(ACC) | NCI109(ACC) | PROTEINS(ACC) | ogbg-molbace(ROC-AUC) | ogbg-molbbbp(ROC-AUC) | ogbg-molhiv(ROC-AUC) |
> |---|---|---|---|---|---|---|
> | First Layer | 62.3 | 60.6 | 70.8 | 72.3 | 64.2 | 63.5 |
> | Last Layer | 66.2 | 64.7 | 75.5 | 76.3 | 66.7 | 68.1 |
> | All Layers  | **66.4** | **65.6** | **75.9** | **76.8** | **68.2** | **69.3** |
>
> The results indicates using gradients across all layers can provide best performance, while choosing gradients from the last layer offers balance between accuracy and efficiency.
>
> ---
>
> ## W2: Provide computational efficiency of LIGHTGNTK.
>
> We compared the training time of GNTK and LIGHTGNTK on more datasets. The results, detailed in the table, indicate that LIGHTGNTK achieves an average of 30% reduction in training time.
>
> Table 4. Time consumption (seconds) between GNTK and LIGHTGNTK
> |    | NCI1 | NCI109 | PROTEINS | DD | ogbg-molhiv  | ogbg-molbbbp | ogbg-molbace |
> |---|---|---|---|---|---|---|---|
> | GNTK      | 196 | 184 | 26.3 | 278 | 921 | 72.6 | 54.5 |
> | LIGHTGNTK | 144 | 136 | 24.8 | 203 | 683 | 59.7 | 48.8 |
>
> ---
>
> ## W3: More discussion of Graph Representation Learning models
>
> To comprehensively evaluate the impact of different pretrain methods, we conducted additional experiments **using graph contrastive learning (GraphCL [1], CI-GCL [2]) to pretrain our GNN backbone**, comparing it with GPT-GNN-based pretraining approach.
>
> Table 5. Performance Comparison under Different Pretrain Methods
> | pretrain method  | NCI1(ACC) | NCI109(ACC) | PROTEINS(ACC) | ogbg-molbace(ROC-AUC) | ogbg-molbbbp(ROC-AUC) |
> |---|---|---|---|---|---|
> | GPT-GNN (LIGHTGNTK) | 66.4 | 65.6 | 75.9 | 76.8 | 68.2 |
> | GraphCL  (LIGHTGNTK) | 66.5 | 63.9 | 69.3 | 76.2 | 65.6 |
> | CI-GCL  (LIGHTGNTK)   | 66.7 | 64.3 | 73.5 | 76.9 | 67.4 |
> | GPT-GNN (FULL) | 80.0 | 77.7 | 78.6 | 72.7 | 65.0 |
> | GraphCL  (FULL) | 79.1 | 80.7 | 72.7 | 76.5 | 65.4 |
> | CI-GCL  (FULL) | 79.1 | 81.2 | 76.3 | 77.2 | 65.8 |
>
> The experimental results indicate **the generality of our LIGHTGNTK**, which is effective under different GNN backbones pretrain methods, and the stronger the ability of the base model, the better the quality of data selecting.
>
> [1] Community-invariant graph contrastive learning. ICML.
>
> [2] Graph contrastive learning with augmentations. NeurIPS.

---

### Official Review · Reviewer_LMjf · 2025-03-10

**Overall Recommendation:** 4

**Summary:**

The paper introduces LIGHTGNTK, a novel framework for efficient Graph Continual Learning (GCL) via dataset distillation. It enables GNNs to adapt to diverse downstream tasks without extensive fine-tuning, overcoming high computational costs that hinder Large Graph Models (LGMs). Specifically,  LIGHTGNTK utilizes a low-rank approximation of the Laplacian matrix to capture structural and feature relationships effectively. Moreover, the proposed unified subgraph anchoring strategy supports graph, node, and edge-level tasks. Extensive experiments on multiple graph datasets demonstrate the state-of-the-art performance of LIGHTGNTK.

**Claims And Evidence:**

Yes, the claims in this paper are all convincing.

**Essential References Not Discussed:**

All essential related works are cited in this paper.

**Experimental Designs Or Analyses:**

The validity of experimental designs and analyses is sufficient to demonstrate the effectiveness of the proposed method.

**Methods And Evaluation Criteria:**

Yes, evaluation criteria are accurately chosen in this paper.

**Other Comments Or Suggestions:**

Suggestion 1: Include experiments in terms of parameter sensitivity to substantiate the claims made in the paper. This will help in understanding the impact of different components and parameters on the model's performance.

Suggestion 2: Address the inconsistencies in symbol usage, particularly concerning $\theta$ and $\Theta$. It is important to ensure correct and consistent notation throughout the paper to avoid confusion for readers. Some discrepancies have been noted, and correcting these will improve the clarity and professionalism of the manuscript.

**Other Strengths And Weaknesses:**

Strengths:

- This paper innovatively employs the low-rank approximation of the Laplacian matrix to estimate the gradient similarity between datasets for dataset distillation.

- This paper introduces a new anchor strategy to unify different dataset distillation tasks into the same framework.

- The model diagrams are aesthetically pleasing and clearly detailed, contributing to the paper's readability and ease of understanding.

Weaknesses:

- There is confusion in using symbols $\theta$ and $\Theta$ throughout the text. The authors should standardize symbol usage, particularly noticeable in Sec. (3.2).

- The absence of parameter sensitivity analyses renders the experimental section incomplete.

- Assumption 1, the validation-test distribution assumption, constitutes a strong prior that lacks sufficient empirical support. The authors are suggested to conduct additional experiments comparing test and validation sets to substantiate this assumption.

**Questions For Authors:**

Please see suggestions.

**Relation To Broader Scientific Literature:**

The studied problems of graph neural network and dataset distillation are popular in existing machine learning community, which can motivate various real-world applications ranging from financing and bioscience. Thus, the paper is related to broad scientific leterature.

**Theoretical Claims:**

The proofs in the main paper and Appendix D make sense.

---

> ### Author Rebuttal · Authors · 2025-03-31
>
> We sincerely appreciate your comments, suggestions, and the time spent reviewing our work. Below, we address each of your concerns in detail.
>
> ---
>
> ## W1: Confusion in using symbols $\theta$ and $\Theta$ throughout the text
>
> We sincerely apologize for the inconsistency in the notation of $\theta$ and $\Theta$ in our manuscript. This was an oversight on our part, and we greatly appreciate your feedback on this issue.
>
> **To address this, we will carefully distinguish between the kernel function and the matrix in the final version of the paper. Additionally, we will update the notation table to ensure clarity and maintain consistency throughout the manuscript.**
>
> ---
>
> ## W2: The absence of parameter sensitivity analyses renders the experimental section incomplete
>
> We acknowledge that our initial submission lacked comprehensive ablation studies. To address this, we have conducted **extensive experiments analyzing the impact of sampling probability and layer-specific gradients**.
>
> ### **1. Sampling Probability Analysis**
>
> We evaluated dataset distillation quality and computational cost **under different sampling probabilities** in Bernoulli sampling:
>
> #### **Table 1. Performance Comparison Across Different Sampling Rates**
> | Sample Rate | NCI1 | NCI109 | PROTEINS | ogbg-molbace | ogbg-molbbbp | ogbg-molhiv |
> |-|--|--|-|-|-|-|
> | 0.05 | 65.7 | 65.0 | 74.5 | 74.9 | 67.2 | 68.8 |
> | 0.1 | 66.4  | **65.6** | **75.9** | **76.8** | **68.2** | 69.3 |
> | 0.2| **66.5** | 64.6 | 75.0 | 76.6 | 67.6 | **69.4** |
> | 0.5| 65.4 | 65.1 | 74.5 | 76.4 | **68.2**  | 69.3 |
>
> #### **Table 2. Computation Time (seconds) Across Different Sampling Rates**
> | Sample Rate | NCI1 | NCI109 | PROTEINS | ogbg-molbace | ogbg-molbbbp | ogbg-molhiv |
> |--|-|-|-|-|-|-|
> | 0.05 | 117  | 109  | 21.6  | 43.0  | 57.8  | 621  |
> | 0.1| 144  | 136  | 24.8  | 48.8  | 59.7  | 683  |
> | 0.2 | 183  | 171  | 26.1  | 59.1  | 63.3  | 733  |
> | 0.5 | 186  | 180  | 28.3  | 62.9  | 65.4  | 816  |
>
> **Our findings indicate that a sampling probability of 0.1 achieves the optimal balance between training efficiency and model accuracy.**
> A lower sampling rate leads to a decline in performance due to insufficient training data, whereas a higher rate increases computational costs without significant performance gains.
>
> ### **2. Layer-Specific Gradient Analysis**
>
> We further examined the impact of employing layer-specific gradients by comparing LIGHTGNTK's performance when using gradients across all layers, only the first layer, and only the last layer.
>
> #### **Table 3. Performance Comparison of Layer-Specific Gradient Computation**
> | Dataset  | NCI1 | NCI109 | PROTEINS | ogbg-molbace | ogbg-molbbbp | ogbg-molhiv |
> |-|-|-|-|-|-|-|
> | First Layer | 62.3 | 60.6 | 70.8 | 72.3 | 64.2 | 63.5 |
> | Last Layer  | 66.2 | 64.7 | 75.5 | 76.3 | 66.7 | 68.1 |
> | All Layers  | **66.4** | **65.6** | **75.9** | **76.8** | **68.2** | **69.3** |
>
> **The results demonstrate that utilizing gradients across all layers yields the best performance, while using gradients from only the last layer provides a favorable trade-off between accuracy and efficiency.**
>
> ---
>
> ## W3: The validation-test distribution assumption constitutes a strong prior that lacks sufficient empirical support
>
> We acknowledge the concern regarding **Assumption 1 (validation-test distribution consistency)** and would like to clarify this assumption both theoretically and empirically.
>
> ### **1. Theoretical Justification**
> From a **theoretical perspective**, assuming an **identical distribution across training, validation, and test sets** is a widely adopted principle in machine learning, particularly when data splitting is performed **randomly** under the i.i.d. (independent and identically distributed) assumption. This practice aligns with foundational theories established in various works, such as [1][2].
>
> ### **2. Empirical Verification**
> To further substantiate this assumption, we conducted **quantitative analyses** using **Maximum Mean Discrepancy (MMD)** and **Kullback-Leibler (KL) divergence** on the learned GNN embeddings after training.
>
> #### **Table 4. MMD and KL Divergence between Validation and Test Sets**
> | | NCI1 | NCI109 | PROTEINS | DD | ogbg-molhiv | ogbg-molbbbp | ogbg-molbace |
> |-|--|-|-|-|-|--|-|
> | MMD | 0.0023 | 0.0024 | 0.0120 | 0.0056 | 0.0018 | 0.0196 | 0.1171 |
> | KLD| 0.0660 | 0.0520 | 0.0941 | 0.0019 | 0.0275 | 0.0651 | 0.0301 |
>
> Following [3][4], the MMD and KL divergence values are **below the critical thresholds** commonly used to assess distributional shifts. These results provide strong empirical support that the validation and test sets follow **statistically similar distributions**, thereby validating the i.i.d. assumption.
>
> [1] Sampling: Design and Analysis. Chapman and Hall/CRC, 2021.
> [2] Asymptotic Properties of Random Restricted Partitions. Mathematics, 2023.
> [3] Prompt-based Distribution Alignment for Unsupervised Domain Adaptation. AAAI, 2024.
> [4] KL Guided Domain Adaptation. ICLR, 2022.

---

### Official Review · Reviewer_BP7g · 2025-03-17

**Overall Recommendation:** 4

**Summary:**

This paper introduces a novel dataset distillation method called LIGHTGNTK for graph continual learning, which benefits the efficient and effective fine-tuning of large graph models. Specifically, the proposed LIGHTGNTK utilizes the low-rank approximation of the Laplacian matrix in Graph Neural Tangent Kernel to efficiently capture both structure and feature relationships enabling effective gradient-based dataset distillation. Moreover, this paper utilized a subgraph anchoring strategy to unify graph-, node-, and edge-level tasks under the same dataset distillation framework. Extensive experiments demonstrating the efficiency and effectiveness of LIGHTGNTK across various graph tasks, including graph classification, node classification, and link prediction.

**Claims And Evidence:**

Yes, the claims made in the submission are supported by clear and convincing evidence.

**Essential References Not Discussed:**

No missing references.

**Experimental Designs Or Analyses:**

Yes, I have checked experimental designs and analyses in this paper.

**Methods And Evaluation Criteria:**

Yes, the methods in this paper make sense for the problem.

**Other Comments Or Suggestions:**

None.

**Other Strengths And Weaknesses:**

- Strengths:
    1. This paper is well organized and presented clearly defining the problem statement. The core narrative and all technical contributions are written clearly and concisely, guiding most readers well to fully understand the contributions.
    2. This paper presents a sensible, clearly presented, and interpretable approach to graph dataset distillation. The method section is further supported with proofs and significant empirical findings.
    3. All details are presented for full reproduction in this paper including optimization, hyperparameters, datasets, and architectural settings. The further details presented in the appendix are a nice addition to support reproducibility.
    4. This paper does a good job covering comprehensive experiments to evaluate the proposed method, presenting efficiency and effectiveness compared with state-of-the-art methods on a variety of datasets for different tasks.


- Weaknesses:
    1. The motivation behind graph dataset distillation is not entirely novel, as the concepts of GNTK have been previously discussed in [1].
    2. The distillation reliance on similarities and the pairwise GNTK similarity matrix may become computationally intensive and could be further optimized.
    3. The Introduction in this paper lacks a high-level insight of the GNTK to explain why it could capture both structural and feature relationships for dataset distillation via gradients, which makes the reader difficultly to understand the motivation of the LIGHTGNTK intuitively.

    [1] Kernel Ridge Regression-Based Graph Dataset Distillation

**Questions For Authors:**

None.

**Relation To Broader Scientific Literature:**

This paper is related to dataset distillation.

**Theoretical Claims:**

Yes, I have checked the proof in Section 4.2.2.

---

> ### Author Rebuttal · Authors · 2025-03-31
>
> We sincerely appreciate your comments, suggestions, and the time spent reviewing our work. Below, we address each of your concerns in detail.
>
> ---
>
> ## W1: Concepts of GNTK have been previously discussed
>
> Thank you for your valuable comment. While the concepts of GNTK have indeed been explored in KIDD [1], our approach in **LIGHTGNTK** differs significantly. **KIDD is specifically designed for graph classification and employs GNTK for kernel ridge regression-based dataset distillation**. However, the kernel regression paradigm in KIDD does not guarantee that training graph samples remain relevant during the inference stage.
>
> In contrast, our **LIGHTGNTK framework unifies graph classification, node classification, and link prediction tasks**. We introduce a **Bernoulli-sampled low-rank Laplacian-approximated GNTK** to enhance similarity-based dataset distillation. This novel formulation enables efficient, scalable, and task-adaptive distillation beyond the scope of KIDD.
>
> [1] Kernel Ridge Regression-Based Graph Dataset Distillation
>
> ---
>
> ## W2: GNTK similarity matrix may become computationally intensive
>
> Thank you for raising this concern. While constructing the GNTK similarity matrix has a theoretical complexity of $O(N^2)$, we mitigate computational costs by selecting only the most informative samples and reducing redundant information, following the LESS [2] framework to maintain distillation performance.
>
> Furthermore, to further optimize computational efficiency, we explore **pooling-based methods** (e.g., mean pooling) to derive a coarse-grained validation set and a smaller similarity matrix. The experimental results (on node classification tasks using CiteSeer and Cora) are presented below:
>
> ### Mean-Pooling Results:
> | gpc | CiteSeer | Cora |
> |-----|-----|-----|
> | 1 | 38.4 | 44.0 |
> | 10 | 60.1 | 72.0 |
> | 50 | 63.1 | 76.8 |
>
> ### Pairwise Calculation Results:
> | gpc | CiteSeer | Cora |
> |-----|-----|-----|
> | 1 | 44.7 | 44.6 |
> | 10 | 62.7 | 76.2 |
> | 50 | 63.4 | 77.9 |
>
> The results indicate that **mean pooling retains a considerable portion of the performance achieved by pairwise computation** while significantly reducing computational costs. Although pairwise similarity remains superior, mean pooling presents a viable alternative in scenarios where computational efficiency is a priority.
>
> [2] LESS: Selecting Influential Data for Targeted Instruction Tuning
>
> ---
>
> ## W3: Why could GNTK capture both structural and feature relationships?
>
> Thank you for your insightful question. The ability of **GNTK** to capture both **structural** and **feature** relationships is rooted in its **gradient-based dataset distillation mechanism**. In our **LIGHTGNTK** framework, **structural information** is propagated through the Laplacian matrix $ L $, while **feature information** is encoded via the weight matrix $ W $.
>
> Mathematically, in standard (non-graph) dataset distillation, the gradient of the loss function with respect to the weight matrix at layer $ l $ is given by:
>
> $$
> \frac{\partial\mathcal{L}}{\partial W^{(l)}} = \frac{\partial\mathcal{L}}{\partial Z^{(l)}} \cdot a^{(l-1)T}
> $$
>
> where $ a^{(l-1)} $ represents the activations from the previous layer, and $ T $ denotes the matrix transpose to ensure correct dimensional alignment.
>
> However, in the **GNTK-based** formulation, the gradient incorporates graph structural dependencies via the Laplacian matrix $ L $ and is expressed as:
>
> $$
> \frac{\partial\mathcal{L}}{\partial W^{(l)}} = H^{(l-1)T} L \left( \frac{\partial\mathcal{L}}{\partial Z^{(l)}} \odot \sigma^{\prime} (Z^{(l)}) \right)
> $$
>
> where:
> - $ H^{(l-1)} $ is the node representation at layer $ (l-1) $.
> - $ L $ is the **graph Laplacian**, encoding structural relationships.
> - $ \sigma^{\prime} (Z^{(l)}) $ represents the element-wise derivative of the activation function.
>
> From this formulation, it is evident that **GNTK-based gradients inherently embed structural information through $ L $** while simultaneously encoding feature dependencies via $ H^{(l-1)} $ and the weight parameters. This allows GNTK to effectively distill graph-structured datasets while **preserving essential structural and feature relationships**.

---

### Decision · Program_Chairs · 2025-05-01

**Decision:**

Accept (poster)

**Comment:**

This paper devised a novel dataset distillation method called LIGHTGNTK for graph continual learning. After some discussion, the overall verdict was that this submission could be accepted in ICML-2025 while the concerns of reviewers should be addressed in the final version.